# Ascorbate–Glutathione Oxidant Scavengers, Metabolome Analysis and Adaptation Mechanisms of Ion Exclusion in Sorghum under Salt Stress

**DOI:** 10.3390/ijms222413249

**Published:** 2021-12-09

**Authors:** Himani Punia, Jayanti Tokas, Anurag Malik, Andrzej Bajguz, Mohamed A. El-Sheikh, Parvaiz Ahmad

**Affiliations:** 1Department of Biochemistry, College of Basic Sciences and Humanities, CCS Haryana Agricultural University, Hisar 125004, Haryana, India; jiyaccshau@gmail.com; 2Department of Seed Science and Technology, College of Agriculture, CCS Haryana Agricultural University, Hisar 125004, Haryana, India; 3Faculty of Biology, University of Bialystok, Ciolkowskiego 1J, 15-245 Bialystok, Poland; abajguz@uwb.edu.pl; 4Botany and Microbiology Department, College of Science, King Saud University, P.O. Box 2455, Riyadh 11451, Saudi Arabia; melsheikh@ksu.edu.sa (M.A.E.-S.); parvaizbot@yahoo.com (P.A.); 5Department of Botany, Goverment Degree College, Pulwama 192301, Jammu and Kashmir, India

**Keywords:** antioxidants, ion transporters, oxidative stress, proline, reactive oxygen species (ROS), salinity, sorghum

## Abstract

Salt stress is one of the major significant restrictions that hamper plant development and agriculture ecosystems worldwide. Novel climate-adapted cultivars and stress tolerance-enhancing molecules are increasingly appreciated to mitigate the detrimental impacts of adverse stressful conditions. Sorghum is a valuable source of food and a potential model for exploring and understanding salt stress dynamics in cereals and for gaining a better understanding of their physiological pathways. Herein, we evaluate the antioxidant scavengers, photosynthetic regulation, and molecular mechanism of ion exclusion transporters in sorghum genotypes under saline conditions. A pot experiment was conducted in two sorghum genotypes viz. SSG 59-3 and PC-5 in a climate-controlled greenhouse under different salt concentrations (60, 80, 100, and 120 mM NaCl). Salinity drastically affected the photosynthetic machinery by reducing the accumulation of chlorophyll pigments and carotenoids. SSG 59-3 alleviated the adverse effects of salinity by suppressing oxidative stress (H_2_O_2_) and stimulating enzymatic and non-enzymatic antioxidant activities (SOD, APX, CAT, POD, GR, GST, DHAR, MDHAR, GSH, ASC, proline, GB), as well as protecting cell membrane integrity (MDA, electrolyte leakage). Salinity also influenced Na^+^ ion efflux and maintained a lower cytosolic Na^+^/K^+^ ratio via the concomitant upregulation of *SbSOS1*, *SbSOS2*, and *SbNHX-2* and *SbV-Ppase-II* ion transporter genes in sorghum genotypes. Overall, these results suggest that Na^+^ ions were retained and detoxified, and less stress impact was observed in mature and younger leaves. Based on the above, we deciphered that SSG 59-3 performed better by retaining higher plant water status, photosynthetic assimilates and antioxidant potential, and the upregulation of ion transporter genes and may be utilized in the development of resistant sorghum lines in saline regions.

## 1. Introduction 

Salt stress is one of the significant abiotic stresses, drastically affecting global agricultural productivity [1,2]. Approximately 23% of the cultivated land is salt-affected, which is comprised of approximately 3.6100 ha [3]. The annual loss caused by soil salinization in agricultural productivity is estimated to be USD 31 million, and the production potential of up to 46 million ha per year. Soil salinization alone has rendered significant chunks of land unproductive or less productive [4]. It is especially concerning since urbanization is shifting farming into more or less arid terrain, 30% of the cultivated lands would become unproductive due to soil salinization. The world’s food demand is expected to rise by 70% by 2050 to feed nearly 9.7 billion people, requiring agricultural productivity gains on smaller land areas and lower water resources [5].

Salinity causes multiple responses in plants, mainly physiological, morphological, biochemical, and molecular alterations [6]. It induces ion imbalance, leading to ion toxicity, osmotic stress, generation of reactive oxygen production (ROS), cellular damage, oxidative damage of membrane lipids, proteins, and nucleic acids [7]. The retention of excess sodium ions in proteins competes with potassium ions, inhibiting metabolic enzymes and protein synthesis [8]. Concentrations of NaCl higher outside the roots impede water absorption, making water extraction more difficult for plants leading to osmotic stress [9]. High salt levels in the leaves induce stomata shutting, electron transport impairment, reducing photosynthetic activity and productivity [10,11,12].

Sorghum (*Sorghum bicolor* (L.) Moench) belongs to the family *Poaceae* and is physiologically classified as C_4_ plants and ranked fifth among the top five economically valuable cereal crops globally [13]. It is grown worldwide in over 41.14 million hectares of area and accounts for 58.72 million metric tons of grains with an average yield of 1.43 metric tons per hectare [14]. Sorghum plays a vital role in global food production and is a dietary staple for more than 500 million people from several rural communities and food-insecure people in many parts of the world [15]. It has a wide range of adaptability under diverse ecological conditions, was moderately drought tolerant, and was highly biomass productive [16,17]. Sorghum has higher resilience to high temperatures, salinity, and drought conditions in the prevailing climate change paradigm, the crop is expected to become more crucial, making it an excellent food and feed resource [13]. Sorghum possesses a diverse range of morphological, anatomical, and physiological characteristics that enables it to survive under stress conditions [18].

Plants emerging in saline soils regulate ion accumulation so that ion toxicity is avoided and appropriate osmotic solutes are retained [19]. This osmotic adjustment maintains the volume and turgor pressure of the cells and organelles of the growing plant. The roots of the constantly transpiring plants must exclude the excess of cations and anions in saline soils that also depends on the type of root and plant species to avoid ion toxicity [20]. Under an excess of Na^+^ ions, plants adapt by triggering a complex network of defense mechanisms that allow them to maintain cellular and ion homeostasis [21]. The ROS generated are scavenged by an efficient network of enzymatic and non-enzymatic antioxidants [17] that includes an array of the ascorbate–glutathione pool like superoxide dismutase (SOD), catalase (CAT), glutathione, ascorbic acid, phenolic, and tocopherols [7,22]; their up-regulation helps in alleviating salinity-induced oxidative damage [23,24]. For example, the intake, partitioning, or removal of ions from specialized cells under salinity is necessary for normal growth and development [23].

Roots are the first organs experiencing high Na^+^ ions, which display a more significant reduction in growth than shoots. Interestingly, age-related differences in responses in plant cells, tissues, and organs have been observed [25]. However, variations were considerably less studied in oxidative stress and antioxidant protection in the different organs and stages of development. Sorghum is an economically important climate-resilient cereal crop with wider ecological adaptability. Therefore, this work aims to elucidate the effects of salinity on antioxidant potential, oxidative stress, and the expression of stress-inducible genes in roots and young and mature leaves harvested from vegetative and physiological maturity in salt-treated and non-treated sorghum genotypes.

## 2. Materials and Methods

### 2.1. Experimental Design

The experiment was conducted during the *kharif* season (2017–2018, 2018–2019, and 2019–2020) in the screen house of the Department of Biochemistry, CCS Haryana Agricultural University, Hisar, Haryana, India. A preliminary experiment was performed wherein 23 sorghum genotypes were screened for salinity tolerance based on germination studies (Appendix A). Based on the germination results, SSG 59-3 was identified as expressing a salt-tolerant genotype and PC-5 a salt-susceptible genotype, which was used for further biochemical and molecular studies (Figure 1).

### 2.2. Plant Material and Growth Conditions

Seeds of the two sorghum genotypes viz. SSG 59-3 (salt-tolerant) and PC-5 (salt-susceptible) (Appendix A) were obtained from the Forage Section, Department of Genetics and Plant Breeding, CCS Haryana Agricultural University Hisar, India. A screen-house experiment was conducted with standard conventional cultivation practices, where sorghum seeds were sown directly into plastic pots lined with polyethylene bags filled with 10 kg sandy loam soil at a 2–3 cm depth. The experiment was set up as a completely randomized design (CRD) with ten plants per treatment (five plants per replicate). The pots were saturated with desired salinity levels, i.e., 60, 80, 100, and 120 mM NaCl (3:1 chloride dominated salinity) maintained at 25/20 °C under a 14 h light/10 h dark cycle. Controls were run simultaneously for two genotypes. All pots were rinsed with an equal volume of water and nutrient solution per the recommended package of practices (POP). The physico-chemical properties of the soil used were determined before sowing (Appendix A). The experiment was conducted using three biological replicates.

### 2.3. Treatments

Saline solutions of required molarity were prepared by mixing NaCl, CaCl_2_·2H_2_O, MgCl_2_·6H_2_O, and MgSO_4_·7H_2_O in the appropriate amount (Appendix A). The soil’s electrical conductivity was checked at an interval of 15 days and at the sampling stage to maintain the desired salt concentration. Three replications of each treatment were taken.

### 2.4. Raising of the Crop

The seeds (10) of uniform size were selected and surface sterilized with 0.01% mercuric chloride (HgCl_2_) solution to prevent the fungal attack and then thoroughly washed with distilled water and air-dried before sowing. The seeds were sown at a depth of 5 cm, and after seedling emergence, thinning was carried out for up to five seedlings per pot. The studies complied with relevant institutional, national, and international guidelines and legislation.

### 2.5. Sampling

Samples were collected at the vegetative stage (35 days after sowing, DAS) (Appendix A) and physiological maturity (95 days after sowing, DAS) (Appendix A), immediately frozen in liquid nitrogen, and stored at −80 °C until needed for further biochemical and molecular analysis. The morpho-physiological traits were determined from the flag leaf. The dry weight (DW) was measured after drying the tissues for five days at 70 °C. The mRNA expression profiling of genes was carried at the early vegetative stage. Data on yield and yield attributes were recorded at the maturity stage.

### 2.6. Physiological Indices

The fresh weight of samples was determined from one whole plant, and the sample was kept in a hot air oven at 70 °C until the constant weight was achieved to determine the dry weight per plant from each replication. Root and shoot length was measured from the base to the top growing bud at the vegetative stage and from the bottom to the topmost portion of the plant, excluding arrow length, at maturity.

Relative water content (RWC) was measured from fresh leaf samples, excised, weighed immediately, and placed in distilled water at a constant temperature in diffused light for six hours. Then, leaf discs were dried in an oven at 80 °C for 72 h for dry weight was recorded and calculated as [26]:$$\mathrm{Relative}\;\mathrm{water}\;\mathrm{content}\;\left(\%\right)=\frac{\mathrm{Fresh}\;\mathrm{weight}-\mathrm{Dry}\;\mathrm{weight}}{\mathrm{Fully}\;\mathrm{turgid}\;\mathrm{weight}-\mathrm{Dry}\;\mathrm{weight}}\times 100$$

The osmotic potential (OP) was determined using a psychrometric technique (Model 5199-B Vapor Pressure Osmometer, Wescor Inc. Logan, UT, USA) in the third leaf from the top. The reading (mmole kg^−1^) displayed on the osmometer was recorded using osmolarity reference standards of sodium chloride.

The photochemical quantum yield (F_v_/F_m_) was recorded in intact plants using a chlorophyll fluorometer (OS-30p, Opti-Science, Inc., Hudson, OH, USA) at mid-day. The initial (F_0_) and maximum (F_m_) fluorescence were recorded, and variable fluorescence (F_v_) derived by subtracting F_0_ from F_m_.

The chlorophyll stability index (CSI) was estimated as the method described by [27]. Chlorophyll a, chlorophyll b, and total chlorophyll estimation was conducted by incubating 50 mg of leaf material in 10 mL of DMSO for 3 h at 60 °C and other sets of the same leaf were heated at 32 °C for 1 h in a water bath. After cooling, 10 mL of DMSO was added, and then the absorbance of the solvent was recorded at 663 and 645 nm. The CSI was calculated using the formula given below:$$\mathrm{CSI}\;\left(\%\right)=1-\frac{\left[\mathrm{Total}\;\mathrm{chlorophyll}\;\mathrm{of}\;\mathrm{heated}\;\mathrm{samples}\right]}{\left[\mathrm{Total}\;\mathrm{chlorophyll}\;\mathrm{of}\;\mathrm{non}-\mathrm{heated}\;\mathrm{sample}\right]}\times 100$$

### 2.7. Na^+^: K^+^ Determination

Na^+^ and K^+^ content were determined in 50 mg of dried and well-ground plant material. Samples were digested in concentrated H_2_SO_4_/ HClO_4_ (9:1), and the clear supernatant was analyzed by mass spectrometry (ICP-MS, Finnigan Element XR, Scientific, Bremen, Germany).

### 2.8. Ascorbate–Glutathione Redox Pool

The antioxidative enzymes were determined in a homogenate of 1 g (FW) of leaf and root tissues, prepared in 5 mL of 100 mM sodium phosphate buffer (pH 7.5) containing 0.25% (*v*/*v*) Triton X-100, 10% (*w*/*v*) polyvinylpyrrolidone, and 1 mM phenylmethylsulfonyl fluoride. Superoxide dismutase (SOD; EC 1.15.1.1) activity was determined by measuring the inhibition of NBT (nitroblue tetrazolium) reduction at 560 nm [28]. Catalase (CAT; EC 1.11.1.6) activity was assayed by monitoring the decomposition of H_2_O_2_ at 240 nm [29]. Peroxidase (POD; E.C. 1.11.1.7) activity was determined by the oxidation of pyrogallol (ε = 2.47 mM^−1^ cm^−1^) [30]. Ascorbate peroxidase (APX; EC 1.11.1.11) assay was based on the spectrophotometric monitoring of ascorbic acid oxidation [ε = 2.8 mM cm^−1^] [31]. Glutathione peroxidase (GR; EC 1.11.1.9) was assayed by monitoring non-enzymatic oxidation of NADPH [ε = 6.22 mM^−1^ cm^−1^] [32]. Glutathione reductase (GR; EC 1.6.4.2) was assayed by monitoring the oxidization of 1 mM NADPH per min (ε = 6.22 mM cm^−1^) [33]. Monodehydroascorbate reductase (MDHAR; EC 1.6.5.4) was assayed spectrophotometrically by following the absorbance decrease at 340 nm due to NADH oxidation [ε = 6.2 mM^−1^ cm^−1^] [34]. The activity of dehydroascorbate reductase (DHAR; EC 1.8.5.1) was measured corresponding to 1 nmol of ascorbate produced during the reaction [ε = 14 mM^−1^ cm^−1^] [31]. The soluble protein was extracted by homogenizing 100 mg fresh tissue as per Lowry et al. [35] to determine the specific activity of the enzymes.

### 2.9. Antioxidant Molecules

Ascorbic acid (ASC) was estimated by homogenizing 500 mg of a sample tissue in 5% (*w*/*v*) metaphosphoric acid [36]. The ascorbic acid concentration was determined at 530 nm using ascorbic acid’s standard curve (10–100 µg). The level of oxidized (GSSG) glutathione, reduced glutathione (GSH), and total glutathione (GSH + GSSG) was estimated by homogenizing fresh sample tissue in 5% (*w*/*v*) sulphosalicylic acid [37]. The reduction rate of 5, 5′- dithiobis-(2-nitrobenzoic acid) [DTNB] was monitored at 412 nm. The carotenoid content was extracted in DMSO, and absorbance was read at 480, 645, and 663 nm [38].

### 2.10. Total Antioxidant Capacity

One hundred milligrams of fresh plant tissue was homogenized in liquid nitrogen and extracted in 2 mL of ice-cold 80% ethanol [39]. Ferric reducing antioxidant power (FRAP) reagent, constituting 300 mM of acetate buffer (pH 3.6), 0.01 mM 2,4,6-tripirydylo-S-triazine (TPTZ)) dissolved in 0.04 mM of HCl and 20 mM of FeCl_3_·6H_2_O, was mixed with the ethanolic extract, and absorbance was read at 600 nm using Trolox was used as a standard. For DPPH antioxidant activity, a suitable aliquot (3.5 mL) of DPPH solution in methanol was added to 0.1 mL of sorghum extract followed by an incubation period of 40 min at room temperature in the dark, absorbance was recorded at 515 nm and expressed in milligram per gram Trolox equivalents (TE) [40]. ABTS activity was assayed with slight modifications [41]. The ABTS radical cation solution was prepared by mixing seven mM of ABTS and 2.45 mM of potassium persulfate solution followed by 16 h of incubation in a dark room. After incubation, the mixture was diluted with buffer (pH 7.4), and absorbance was read at 734 nm.

### 2.11. Polyphenolic Compounds

Extraction was performed in 1% HCl/methanol (*v*/*v*) for 2 h with shaking, centrifuged, and pooled supernatants. Total phenols were estimated using gallic acid as a standard for quantifying the samples containing phenolic content, and data are expressed as mg/g gallic acid equivalents (GAE) [42]. Total flavonoids were estimated using catechin as a standard [43]. o-dihydroxy phenols were quantified using chlorogenic acid as the standard [44]. Flavanols were calculated using rutin as a standard [45].

### 2.12. Compatible Osmolytes

The proline content in the sample tissue was analyzed by homogenizing 500 mg of tissue in 5 mL of 3% aqueous sulphosalicylic acid, centrifugation at 5000 rpm for 20 min, and extracted using ninhydrin reagent and toluene extraction [46]. The absorbance was read at 520 nm. Glycine betaine (GB) estimation was conducted in finely powdered plant material (500 mg) [47]. The dried material was mechanically shaken with deionized water, the extract was diluted with 2 N sulfuric acids (1:1), centrifuged at 10,000× *g* for 15 min, and absorbance was measured at 365 nm. Total soluble carbohydrates (TSC) were extracted using 2% phenol and concentrated H_2_SO_4,_ and the absorbance of the solution was measured at 490 nm [48].

### 2.13. Oxidative Stress Markers

Malondialdehyde content was assayed in 250 mg of tissue from the control and stressed plants were ground in 2 mL of chilled 1% TCA and centrifuged at 10,000 rpm for 20 min [49]. After centrifugation, supernatant reacted with 20% (*w*/*v*) trichloroacetic acid (TCA) containing 0.5% thiobarbituric acid to produce pinkish-red chromogen thiobarbituric acid-malondialdehyde (TBA-MDA). Absorbance was measured at 600 nm using the extinction coefficient of 155 mM^−1^ cm^−1^. Hydrogen peroxide (H_2_O_2_) concentration was measured by homogenizing one gram of fresh leaf and root tissue in ice-cold 0.1 M of phosphate buffer (pH 7.0), and 40 µL was used in the assay based on the peroxide-mediated oxidation of Fe^2+^, followed by reaction of Fe^2+^ with xylenol orange at 570 nm [50]. The membrane injury index was calculated as the percent proportion of ion leakage into the external aqueous medium to the total ion concentration of the stressed tissue as measured by the external medium EC [51]. Two hundred mg of leaf and root tissues were boiled in deionized water at 27 °C, and the solution’s electrical conductivity (EC) was measured. The membrane injury index was calculated as follows:$$\mathrm{Membrane}\;\mathrm{injury}\;\left(\%\right)=\frac{\mathrm{EC}1}{\mathrm{EC}2}\times 100$$

### 2.14. Polyamines

In fresh tissue, polyamines were homogenized in 5 mL of 5% cold perchloric acid [52]. The supernatant containing the free polyamines was frozen at −20 °C for further benzoylation reaction using 500 µL of extract, 1 mL of 2N NaOH, 10 µL of 99% (*v*/*v*) benzoyl chloride, incubated for 30 min and then saturated with 2 mL NaCl. Benzoylated polyamines were extracted with diethyl ether, incubated at −20 °C for 1 h, and were separated with the HPLC fitted with C_18_ ODS 2 analytical column, which was pre-equilibrated and ran isocratically at a flow rate of 1 mL min^−1^ with acetonitrile: water (50:50 *v*/*v*) for 40 min. The injected volume was 10 µL, and the UV- detector measured the absorbance at 224 nm. Polyamines were quantified in the eluent by comparing the integration areas obtained for the samples with those obtained for the standards, and the results were expressed as nmol g^−1^ fresh weight of the sample.

### 2.15. Quantitative Real-Time PCR (qPCR) Gene Expression Analysis

qPCR analysis was performed on QuantStudio™7 Flex Real-Time PCR (Applied Biosystems, Thermo Fisher Scientific, Waltham, MA, USA) to study the expression of candidate genes. Total RNA was extracted from fresh tissue using Qiagen Plant Total RNA Miniprep Kit (Qiagen, Germantown, MA, USA). The extracted RNA was quantified using Picodrop (Picodrop Ltd., Cambridge, UK). The total RNA concentration was determined by absorbance at 260 nm, and the A260:280 ratio in the range from 1.8–2.0 was used for further analysis. Single-stranded cDNA was synthesized using an iScript cDNA synthesis kit (Bio-Rad Laboratories, Inc., Pleasanton, CA, USA). Three biological and five technical replicates from each sample were used for qPCR quantification analysis. qPCR analysis was performed in a 20 µL reaction volume using Maxima SYBR Green qPCR master mix (2X) (Thermo Scientific) with thermal cycling conditions of 95 °C for 3 min followed by 40 cycles at 95 °C for 30 s, 57 °C for 30 s and 72 °C for 15 s. The expression of the transcripts in control and treated samples were normalized with *Actin-1* and *PP2A* (protein phosphatase 2A subunit A3) as reference genes, and further experiments were performed with *PP2A* as the reference gene. An internal reference gene and a target gene were compared, using the comparative C_T_ method (∆∆C_T_ method) [53], to analyze the expression levels of a target gene and the internal reference genes. Before sequencing, primer pair specificity was checked using a 2% agarose gel electrophoresis. Primer-BLAST software from NCBI was used for designing primers for the genes viz. salt overly sensitive 1 (*SOS1*), salt overly sensitive 2 (*SOS2*), sodium proton antiporter (*NHX-2*), vacuolar proton pyrophosphatase (*VPPase-II*), and CBL interacting serine-threonine protein kinase 24 (*CIPK24*) (Table 1). The primers then underwent customized synthesis by Sigma. The pooled and diluted cDNA samples were used for qPCR. Using the NCBI database BLASTN algorithm, the sequencing amplification products were verified and found to be 100% identical, confirming the qPCR primer pairs.

### 2.16. Yield Components

Plant height was measured from the base to the topmost leaves at the vegetative stage and from the bottom to the top tip of the plant at maturity, including arrow length and expressed in centimeters (cm). The number of leaves per plant was defined as the mean of simply counted leaves from each replication. The number of tillers per plant was defined as the mean of the estimated tiller numbers from each replication. The seed yield per plant (g) was recorded from the seed weight of each plant.

### 2.17. Statistical Analysis

The data were expressed as Mean ± SD (three replicates each). The analysis of variance (ANOVA) was conducted to check the significance of the main effects (genotypes, salinity, and stage) and their interaction on growth indices. Significant differences between the means of parameters (*n* = 8) were determined using Tukey’s test at a 5% level (*p* ≤ 0.05). Statistical analysis was performed using SPSS v25.0 software (SPSS for Windows, Chicago, IL, USA).

## 3. Results

### 3.1. Biomass Accumulation

To assess the effect of high salinity (NaCl) on sorghum growth, plant biomass (fresh and dry weight) under different salt concentrations was measured. With the increasing levels of salt stress from the control to 120 mM NaCl, a significant reduction in fresh and dry plant weight was observed among both genotypes at *p* < 0.05 (Figure 2a,b). At 35 DAS, PC-5 had the maximum reduction (79.6%) in fresh weight while minimum in SSG 59-3 (29.9%) at 100 mM NaCl, while at 120 mM, the decline was higher, 87.6% in PC-5 and 41.9% in SSG 59-3. At two sampling stages, SSG 59-3 had increased biomass production over PC-5. Root length decreased significantly with the increasing concentration of NaCl in all the genotypes at 35 and 95 DAS (Figure 2c). At 35 DAS, a minimum percent reduction was observed in SSG 59-3 (26%) and a higher reduction in PC-5 (69%) at 100 mM NaCl. While at 120 mM, the percent decrease was higher, PC-5 had a maximum reduction of 78%, while SSG 59-3 had only a 36% reduction over their respective controls. A similar trend was noticed for shoot length (Figure 2d). SSG 59-3 (108.6 cm) acquired maximum shoot length compared to PC-5 (48.3 cm). A significant interaction was found at both sampling stages, i.e., 35 and 95 DAS.

### 3.2. Morpho-Physiological Analysis

Relative water content (RWC) of leaves was relatively low under salt stress compared with control conditions in all the sorghum genotypes (Figure 3a). At 35 DAS, the reduction in RWC under salt stress was much higher in PC-5 (25.2%) than SSG 59-3 (7%) at 100 mM NaCl. SSG 59-3 maintained RWC under increasing salt concentrations up to 100 mM NaCl, but at 120 mM, there was a decrease in RWC. The osmotic potential of leaves was measured in terms of –MPa at two developmental stages (Figure 3b). The osmotic potential of the leaf declined progressively with the increasing level of salt from 0 to 120 mM NaCl. At 35 DAS, values of osmotic potential became more pessimistic in PC-5 than SSG 59-3, i.e., −1.13 MPa and −0.71 MPa, respectively, at 120 mM NaCl. At 95 DAS, trends were similar to 35 DAS.

A significant difference in chlorophyll fluorescence/photochemical quantum yield was observed within the genotypes and treatments (Figure 3c). At 35 DAS, SSG 59-3 showed more quantum yield than PC-5. Percent reduction was higher in PC-5, i.e., 54% and 62%, and lower in SSG 59-3, i.e., 20% and 28% at 100 and 120 mM NaCl, respectively. At 120 mM, the reduction in quantum yield was more than 100 mM, suggesting that 100 mM concentration was under the physiological tolerance range of sorghum crop. At physiological maturity (95 DAS), the photochemical quantum yield decreased significantly in all salt treatments. Salt stress significantly reduced the chlorophyll stability index (CSI) (%) of sorghum genotypes at *p* < 0.05 (Figure 3d). At 35 DAS, the reduction was more eminent in the genotype PC-5 (65.3%) trailed by the genotype SSG 59-3 (22.9%). The decline in chlorophyll stability in terms of loss of chlorophyll was higher in PC-5 (85.3%) and lower in SSG 59-3 (30.2%) at 120 mM NaCl, suggesting the tolerance behavior of the SSG 59-3 genotype. PC-5 exhibited the highest percent reduction (81.3%) at 120 mM NaCl at physiological maturity, indicating its susceptibility toward salinity due to severe loss in chlorophyll content. At the same time, SSG 59-3 had a 46.8 percent reduction, indicating its stability toward chlorophyll content.

During the vegetative stage, the photosynthetic rate was low, and hence, PC-5 (52.5%) had a decrease in chlorophyll content as compared to SSG 59-3 (22.5%) at 100 mM (Figure 4a). The analysis of Figure 4b demonstrates that as the NaCl level decreased from the control to 120 mM NaCl, a parallel reduction was recorded in the chlorophyll b content. A considerable difference was observed within the genotypes and treatments. Total chlorophyll content followed a similar pattern (Figure 4c). The overall mean value of chlorophyll a, chlorophyll b, and total chlorophyll content was the maximum at 35 DAS compared to 95 DAS.

### 3.3. Na^+^: K^+^ Determination

High soil salinity causes the over-accumulation of Na+ inside the cell, resulting in ionic imbalances and metabolic toxicity. Na^+^:K^+^ ratio (Figure 5a) increased with applied NaCl concentration in leaves and roots. Na^+^ accumulation was higher in roots with increasing salt concentration, particularly at 120 mM concentration at *p* < 0.05 (Figure 5b).

Roots accumulated a higher concentration of Na^+^ ions with an apparent gradient. In SSG 59-3, due to the strong ion exclusion mechanism, it maintained lower Na^+^ ions and imported more K^+^ ions. In contrast, PC-5 had a higher Na^+^/K^+^ ratio due to the higher accumulation of Na^+^ ions. In accord with trends observed in the Na^+^ content in roots, the Na^+^ in leaves also increased, and K^+^ content decreased with increasing salt stress levels.

### 3.4. ASC/GSH Cycle Components

Mild or severe salinity stress increased SOD activity in leaf and root tissues in a concentration-dependent manner (Figure 6a,b). At the vegetative stage (35 DAS), SSG 59-3 had a percentage increase of 54% in leaves at 100 mM NaCl while the increase was only 18.2% in PC-5. At a higher salt concentration, i.e., at 120 mM NaCl, SOD activity was more pronounced with an increase of 63.9% in SSG 59-3 and 33.2% in PC-5 leaves. A similar trend was found in roots in all the genotypes; however, SOD activity in roots was lower than leaf tissue at both the growth stages. At 35 DAS, the percentage increase was the maximum in SSG 59-3 (22%) and lower in PC-5 (14%) at 100 mM NaCl. Similar results were observed at physiological maturity with a higher percentage increase than the vegetative stage. Both leaves and roots showed differential CAT activity expressed as Units mg^−1^ protein (Figure 6c,d). At 100 mM, SSG 59-3 had a percentage increase of 48.9% while PC-5 had only a 22.6% increase, while at 120 mM NaCl, higher CAT activity was observed with 65.2% in SSG 59-3 and 32.4% in PC-5. Similar results were observed at physiological maturity with a higher percentage increase than the vegetative stage. A similar trend was observed in all genotypes; however, CAT activity in roots was lower than leaf tissue at both growth stages. At 35 DAS, a similar increasing trend of CAT activity was observed in the SSG 59-3 genotype (28%), and the percentage increase was less pronounced in the genotype PC-5 (18%) at 100 mM NaCl. Similar results were observed at 95 DAS. An increase in POX activity was observed in both sorghum genotypes (Figure 6e,f). In leaves at 35 DAS, the increase in POX activity in SSG 59-3 was 58% and 62% at 100 mM NaCl and 120 mM NaCl, whereas PC-5 had a slight increase (22% and 29%). A similar trend in POX activity was observed at 35 DAS with values approaching approximately 4.4-fold in the genotype SSG 59-3 while a 2.9-fold in PC-5 under saline conditions. Roots also had a similar basal level of POX activity that was observed in the leaves at both developmental stages. However, the percentage increase in POX activity was less than the leaves in all the sorghum genotypes under stress conditions. At 100 mM NaCl, the maximum increase in POX activity was found in the salt-tolerant genotype SSG 59-3 (26%) while less in PC-5 (18%) at 35 DAS. A significant increase in specific activity was observed at a higher salt concentration (120 mM NaCl) from 32% in SSG 59-3 to 22% in PC-5. Similar results were noticed at 95 DAS.

At 35 DAS, APX activity increased significantly in SSG 59-3 (55%), whereas in PC-5, the increase (26%) in APX activity was slightly less at 100 mM NaCl (Figure 6g,h). Further increase in salt concentration i.e., at 120 mM NaCl enhanced the APX activity, more in SSG 59-3 (65%) and less in PC-5 (23%). The same trend was observed at 95 DAS, but overall, APX activity was more at 95 DAS than 35 DAS. A pattern similar to the leaves was also observed in roots, but overall activity was more in leaves. At 35 DAS, a similar trend of increased APX activity in roots was observed in all the genotypes. SSG 59-3 (26%) showed maximum APX activity, whereas PC-5 (19%) had less APX activity at 100 mM NaCl. Similar results were observed at 95 DAS, but the mean value was the maximum at 95 DAS compared to 35 DAS. Differential response of GPX activity (Units mg^−1^ protein) showed a significant increase (Figure 7a,b). At 35 DAS, a significant increase in GPX activity was observed in the leaves of the salt-tolerant genotype SSG 59-3 (45%), with a slight increase in the salt-sensitive genotype PC-5 (24%) 100 mM NaCl. At 120 mM NaCl, the GPX activity was higher in SSG 59-3 (58%) as compared to PC-5 (33%). At 95 DAS, a similar trend in the increase in GPX activity was observed with a maximum in SSG59-3 and a minimum in PC-5. A similar trend was found in all genotypes; however, GPX activity was lower than leaf tissue at both the growth stages. At 35 DAS, a similar increasing trend in GPX activity was observed in the SSG 59-3 genotype (30%), and the percentage increase was less pronounced in the genotype PC-5 (16%) at 100 mM NaCl. Similar results were observed at 95 DAS, but overall maximum GPX activity was found at 95 DAS compared to 35 DAS.

Salt stress increased the GR activity in both the tolerant and susceptible sorghum genotypes (Figure 7c,d). However, the increase was found to be higher in tolerant genotypes. In the SSG 59-3 genotype, GR activity increased by 48% and 55% compared to PC-5, where the percent increase was 22% and 33% at 100 mM NaCl and 120 mM NaCl, respectively, at the vegetative stage. In roots, at the vegetative stage, the percent increase in GR activity was 30% and 42% in SSG 59-3 and 16% and 21% in PC-5 at 100 mM NaCl and 120 mM NaCl, respectively. Among the studied genotypes, PC-5 showed the lower, while SSG 59-3 had the higher, GR activity under salinity conditions. MDHAR activity increased significantly at 35 DAS in the sorghum genotypes under salt stress (Figure 7e,f). At 35 DAS, the activity increased by 53% in SSG 9-3, while the increase was less pronounced in PC-5 (28%) at 100 mM NaCl. A more pronounced increase was observed at 120 mM NaCl with a maximum increase in SSG 59-3 (68%) and less in PC-5 (38%). A similar trend of increase in MDHAR activity was observed at 95 DAS, leaves accumulated higher activity than roots at 95 DAS, reaching up to 72% and 79% in SSG 59-3 and 42% and 55% in PC-5 at 100 mM NaCl and 120 mM NaCl, respectively. A similar trend was recorded in the roots of studied genotypes; however, MDHAR activity in the roots was lower than leaf tissue at both the growth stages. Overall, the mean value of specificity activity was more at 95 DAS as compared to 35 DAS. Among the studied genotypes, PC-5 showed the lower, while SSG 59-3 had the higher, DHAR activity under salinity conditions (Figure 7g,h). In SSG 59-3, DHAR activity increased by 61% and 67% compared to PC-5, where the percentage increase was 31% and 45% at 100 mM NaCl and 120 mM NaCl, respectively, at the vegetative stage. In roots, at the vegetative stage, the percentage increase in DHAR activity was 30% and 41% in SSG 59-3 and 23% and 28% in PC-5 at 100 mM NaCl and 120 mM NaCl, respectively. Among both genotypes, PC-5 showed lower, while SSG 59-3 had higher DHAR activity under salinity treatment. Leaves accumulated higher activity than roots at 95 DAS.

### 3.5. Enzymatic Antioxidants

A significant increase in total glutathione content in sorghum tissues was observed at both growth stages (Figure 8a,b). In leaves, SSG 59-3 (54.3%) efficiently had higher glutathione as compared to PC-5 (23.6%) at 100 mM concerning their controls. A further increase in salt concentration had a pronounced effect on glutathione content with 68.5% in SSG 59-3 and 28.9% in PC-5 over their respective controls. Similar results were observed for reduced glutathione content (GSH) (Figure 8c,d). While in oxidized glutathione (GSSG), SSG 59-3 leaves (54.6%) maintained higher GSSG than PC-5 (25.4%) (Figure 8e,f). In roots, the values of total glutathione, GSSG, and GSH were not higher than leaves. A similar increase in glutathione content was observed at 95 DAS, but the overall mean value was higher in leaves in comparison to roots.

Differential response in ascorbic acid (ASC) content was observed in sorghum genotypes (Figure 9a,b). At the vegetative stage, the percent increase was 54.65 in SSG 59-3 while PC-5 had a 21.5% increase at 100 mM NaCl over their respective controls. A significant decline in the carotenoid content of all the genotypes under stress conditions was observed with a higher decrease in the salt-susceptible genotype (PC-5) as compared to the tolerant one (SSG 59-3) at both the developmental stages (Figure 9c). Higher percent reduction was recorded in PC-5, i.e., 40% and 52% and lower percentage reduction in SSG 59-3, i.e., 18% and 24% at 100 mM and 120 mM NaCl, respectively.

### 3.6. Antioxidant Capacity and Polyphenolic Compounds

Antioxidant activities viz. FRAP, DPPH, and ABTS were evaluated to exploit the antioxidant potential of two sorghum genotypes at *p* < 0.05 (Table 2). The red-colored SSG 59-3 grains showed high FRAP and DPPH activities compared to the PC-5. SSG 59-3 exhibited the highest increase in antioxidant activity in the ABTS assay at 43.76 ± 0.23 mg/100 g. The lowest was indicated by white PC-5 grains at 20.24 ± 0.15 mg/100 g. In the FRAP assay, SSG 59-3 exhibited the highest antioxidant activity at 16.64 ± 0.10 mg/g TE, while lower activity was shown by the white PC-5 grains. A significant difference in total phenolic content was observed between the different sorghum varieties. The red pericarp SSG 59-3 had the highest TPC 940.73 ± 6.7 mg/100 g GAE. The TPC over 291.18 ± 3.87 mg/100 g GAE was less in the white pigmented PC-5. Similar results were observed for flavanols and o-hydroxy phenols.

### 3.7. Compatible Solutes

The accumulation of compatible solutes viz. proline (Figure 10a,b) and glycine betaine (Figure 10c,d) increased considerably in leaves and roots of sorghum genotypes under salt stress. For proline, the percentage increase in leaves was the maximum in SSG 59-3 (50%) while PC-5 had the minimum of 23%. For glycine betaine, the percentage increase was the maximum in SSG 59-3 (58%), while PC-5 had 23% at 100 mM. In roots, the glycine betaine level increased significantly, but the accumulation level was less from leaves in all the genotypes. Salt stress resulted in a significant increase in total soluble sugars (TSS) (Figure 10e,f) at *p* < 0.05 level. At 100 mM, SSG 59-3 leaves accumulated 52% TSS compared to PC-5, where the percentage increase was only 31%. A higher concentration of Na^+^ ions had a pronounced effect on TSS, increasing the total soluble sugar content by 68% in SSG 59-3 and 48% in PC-5. While in roots, the values of total soluble sugar content were less as compared to leaves. A similar increase in total soluble sugar was observed at 95 DAS, but the overall mean value was higher in leaves at 95 DAS than 35 DAS.

### 3.8. Oxidative Stress Markers

Under stress conditions, there is a generation of ROS, which disrupts cellular homeostasis and oxidative stress. Hydrogen peroxide (H_2_O_2_) content, malondialdehyde (MDA), and membrane index injury (MII) are the indications of oxidative damage of cellular structure (Figure 11). H_2_O_2_ content significantly increases under high salt concentrations, more particularly in leaves than roots at *p* < 0.05 level (Figure 11a,b). At 35 DAS, H_2_O_2_ content increased significantly in the genotype PC-5 (42%), whereas its level declined in SSG 59-3 (16%) at 100 mM NaCl. Further increase in salt concentration i.e., at 120 mM NaCl, enhanced the H_2_O_2_ content, more in PC-5 (51%) and less in SSG 59-3 (23%). The same trend was observed at 95 DAS, but overall H_2_O_2_ content was more at 95 DAS than 35 DAS. A pattern similar to the leaves was observed in roots, but overall H_2_O_2_ content was more in leaves. Following the trend of H_2_O_2_, MII (Figure 11c,d) and MDA (Figure 11e,f) also increased in leaves and roots, particularly at 120 mM NaCl in PC-5. Thus, oxidative stress markers showed that root tissues suffered the most from the salinity-induced oxidative stress, as compared to leaves. SSG 59-3 accumulated fewer stress markers due to oxidative damage.

### 3.9. Polyamines

During plant maturation the total polyamine concentration remains constant, but under stressed conditions, their level of accumulation is enhanced (Appendix A). A typical chromatogram of benzoylated polyamine standards was separated by a C18 reverse phase HPLC column (Appendix A). Each peak was confirmed by the respective polyamine standards (Appendix A). The byproducts of the benzoyl reaction represented on the left side of the chromatogram were eluted out and separated very clearly from benzoylated polyamines. The naturally occurring amines are putrescine (Put) (Mol. wt. 161.07), spermidine (Spd) (Mol. wt. 145.25), and spermine (Spm) (Mol. wt. 202.34). The amount of individual amine represented at each peak on the chromatogram is 0.05 mM. Table 3 illustrates the levels of accumulation of different polyamines in leaves under different developmental stages.

The changes in polyamines profiles in the flag leaf under salinity are depicted in Table 3. There is an increased accumulation of spermine (Spm) with increasing levels of salinity in both genotypes. However, an elevated basal level was found in SSG 59-3 (39.64 ± 0.34 nmol g^−1^ FW at 100 mM and 45.16 ± 0.64 nmol g^−1^ FW at 120 mM NaCl) at 35 DAS. The level of accumulation of Spm in the salt-susceptible PC-5 genotype (18.59 ± 0.61 nmol g^−1^ FW at 100 mM and 26.57 ± 0.62 nmol g^−1^ FW at 120 mM NaCl) was not higher as compared to the salt-tolerant SSG 59-3. A similar increase in Spm content with increasing salinity levels was also recorded at 95 DAS, but the overall mean values were higher at 35 DAS. There was a gradual increase in endogenous spermidine (Spd) content in both genotypes. At 35 DAS, the SSG 59-3 had higher Spd content with the progression of salt concentration, more particularly at 100 and 120 mM NaCl, i.e., 314.21 ± 1.9 and 357.95 ± 2.0 nmol g^−1^ FW, respectively. In contrast, PC-5 accumulated lower levels of Spd content, i.e., 164.28 ± 1.3 and 189.37 ± 1.4 nmol g^−1^ FW, respectively. With plant maturity, the Spd content gradually decreased. Thus, the overall mean value of Spd content was higher at 35 DAS than 95 DAS. The accumulation of putrescine (Put) followed a similar pattern as that of Spm and Spd. SSG 59-3 efficiently accumulated higher Put levels than PC-5 with increasing salt concentrations, especially at 35 DAS. However, the basal level in the salt-tolerant SSG 59-3 (457.64 ± 2.3 nmol g^−1^ FW at 100 mM and 496.37 ± 2.8 nmol g^−1^ FW at 120 mM NaCl) was more when compared with the salt-susceptible PC-5 (297.69 ± 1.5 nmol g^−1^ FW at 100 mM and 324.12 ± 1.7 nmol g^−1^ FW at 120 mM NaCl). A slight decrease in the mean Put content was observed at 95 DAS.

### 3.10. Gene Expression Analysis

Quantitative RT-PCR expression analysis was conducted to identify and confirm if the pathways of the selected ion transporter genes were involved in providing tolerance to high salinity in sorghum genotypes. RNA integrity of candidate genes viz. *SOS1*, *SOS2*, *NHX-2*, V*-PPase-II,* and *CIPK24* was assessed by horizontal agarose (1.5%) gel electrophoresis, (Figure 12A,B) and single-band samples specified the gene of interest as well as single melting curve peaks (Figure 12C–G). The relative expression was studied under two different salt concentrations, i.e., at 10 and 12 dS m^−1^. *PP2A* was used as the reference gene/internal control for data normalization. These two different salt concentrations were used for the gene expression study, as the significant differences were not observed at 6 and 8 dS m^−1^. In SSG 59-3, the higher expression of *SOS1* (Figure 12H) and *SOS2* (Figure 12I) under high Na^+^ ions indicated its tolerance behavior to exclude the toxic effects of Na^+^ ions, while in PC-5, the up-regulation of these genes was lower. The expression was more at 120 mM NaCl salt concentration. The expression level of *NHX-2* (Figure 12J) and *CIPK24* (Figure 12M) were highly upregulated at 10 dS/m, while vacuolar transporter gene viz. *V-PPase-II* was highly upregulated in roots (Figure 12L) as compared to leaves (Figure 12K) in salt-tolerant genotype. *PP2A* was constitutively expressed stably under all salt treatments. Their upregulation during salinity indicated the antioxidative cellular defense mechanism to combat the toxic effects of salt accumulation inside the cytoplasm. The higher expression of ion transporter genes under high salinity indicates that SSG 59-3 may be utilized as a salt-tolerant crop due to its better genetic and agronomical traits. The detection of salt-tolerant genes could be adding a molecular base for identifying salt-tolerant genotypes in sorghum.

### 3.11. Yield Traits

SSG 59-3 was found to be superior in retaining the maximum number of leaves per plant than other genotypes at different salinity levels (Table 4). Plant height decreased significantly with increasing salinity levels in sorghum genotypes at *p* < 0.05 level (Table 4). At 35 DAS, the percent decline was higher in the salt-susceptible genotype (PC-5; 76.8%) while lower in the salt-tolerant genotype (SSG 59-3; 26%) over their respective controls at 100 mM. While at 120 mM, the percent decline was 34% in SSG 59-3 and 86% in PC-5, but it further increased significantly with the progression of the developmental stage. A similar trend was noticed at 95 DAS, but the mean value of the plant height was higher at 95 DAS than 35 DAS. SSG 59-3 (65 cm) was significantly superior in retaining plant height at different salinity levels, particularly at 100 mM. The interaction between the treatments and the genotypes was found to be significant at both stages. The number of leaves per plant declined with increasing salt concentration (Table 4). SSG 59-3 was significantly superior in retaining the maximum number of leaves per plant than the other genotypes at different salinity levels. At 35 DAS, the percent reduction in leaf number at 100 mM was 40% in PC-5 and 16% in SSG 59-3 over their respective controls. A similar trend was also noticed at the second sampling stage, i.e., 95 DAS. It is concluded, from the observations at both sampling stages, that SSG 59-3 had more leaves per plant than PC-5.

The number of tillers per plant is the yield determining factor. During physiological maturity, the percent decrease in the number of tillers (Table 5) was observed to be higher in PC-5 at 100 and 120 mM, i.e., 19% and 59%, respectively, while in SSG 59-3, the percent decline was less, i.e., 8% and 19% at 100 and 120 mM, respectively. Due to increased biomass and dry matter, the number of tillers per plant was more pronounced at physiological maturity. Results illustrated in Table 4 show that salt stress significantly affected the seed yield per plant in all sorghum genotypes. At 95 DAS, the percent of harvested seeds was less in PC-5 (32.7%) and higher in SSG 59-3 (72.1%) (Table 5). SSG 59-3 was found to be superior in retaining seed yield at different salinity levels.

## 4. Discussion

Soil salinization has emerged as a severe threat affecting crop productivity and the geographical distribution of crop plants [54]. To contribute to our understanding of the biochemical and molecular mechanisms underlying salinity, we imposed sorghum genotypes to different salinity levels and compared the responses in leaves and roots under two different developmental stages. The increased accumulation of fresh biomass in tolerant genotypes may promote protoplasmic components, accelerated cell division, and elongation contributing to the luxury of vegetative development and increased fresh biomass and dry matter production [55]. In contrast, the decrease in biomass might be a reason for higher Na^+^ ions, which results in delayed physiological maturity of the crop [56]. Decreased root and shoot length leads to high neutral detergent fiber content, which reduces in-vitro dry matter digestibility under osmotic stress in sorghum. The decrease in root and shoot length could be due to increased accumulation of Na^+^ ions, which may cause root tip cell damage and decreased essential micronutrient uptake by plants under salt stress [57].

The physiological characteristics of plants might be adversely affected owing to reduced leaf expansion, photosynthetic impairment, premature leaf senescence, and alterations to the protein and pigment structure [58]. The key factors that govern the relationship between plant waters, productivity, and growth during salt stress include relative water content, osmotic potentials, stomatal conductance, and total chlorophyll. RWC is regarded as a plant water status indicator. It is also proposed that high RWCs might enable tolerant genotypes to more efficiently conduct physio-biological activities under stress than susceptible chickpea genotypes [59]. Osmosis in plant cells can act as a mechanism of osmotic adjustment to decrease the cellular osmotic potential and, thus, maintain water absorption and turgor. A decrease in osmotic potential (ψ_s_) under stress conditions has been suggested as playing an essential role in the osmotic adjustment and plant survival mechanism under dry conditions [60,61]. The primary causes of a high osmotic potential under saline conditions include the hydrolysis of biomolecules into smaller molecules such as mono, oligosaccharides, and proline among amino acids. Photochemical quantum yield (F_v_/F_m_) reveals plant capability for abiotic stress tolerations and how the photosynthetic machinery has been affected under these stressful conditions [62,63]. In photosystem II (F_v_/F_m_), the maximum quantum yield decreased significantly, whereas, in saline conditions, non-photochemical quenching (qN) in sorghum increased substantially [63]. Chlorophyll degradation in leaves under high Na^+^ ions may also be accompanied by ROS generation, resulting in pigment–protein complex instability, chlorophyll oxidation, and degeneration of other chloroplastic pigments [2]. A decline in the concentration of K^+^ and Mg^2+^ can lead to the overall drop in chlorophyll concentration under salinity stress concentrations [64]. Salt stress maintained an abnormally high Na^+^/K^+^ ratio, inactivated the enzymes used in the various developmental processes, and inhibited protein synthesis [62]. Another interesting point is that sorghum can also accumulate higher Na^+^ concentration in the roots and limit the transportation of Na^+^ up to shoots under salt stress, which is called salt exclusion [65], a critical salt-tolerance-related process. Growth inhibition in maize organs was correlated linearly with increasing Na^+^ accumulation within tissues [9].

The objective of this study was to establish a correlation between cellular antioxidant mechanisms and salinity-induced alterations in the root and leaf tissues of sorghum genotypes at two different developmental stages. During oxidative stress, comprehensive enzymatic antioxidant properties scavenge excess ROS generation, thereby regulating ROS levels and protecting the crop from oxidative damage [66]. Different antioxidant enzymes, salt concentration, and exposure time are all key attributes in this salinity-induced coping strategy [67]. SOD is the most potent cytoplasmic enzymatic protector, present in all organisms and intracellular compartments that are sensitive to ROS-mediated oxidation [22]. Transgenic tobacco plants overexpressing Cu/Zn-SOD displayed resistance to salt stress and drought [68]. The accumulation of H_2_O_2_ causes the activation of catalase and is seemingly consistent in scavenging enhanced H_2_O_2_ levels [69,70]. POD activity was induced in tolerant genotypes than in sensitive genotypes, indicating that it may play a significant role in rapidly eliminating H_2_O_2_ in tolerant genotypes under salinity. In *A. lividus*, a dramatic reduction in POD activity and enhanced heat shock proteins was observed under salinity stress [71]. Higher activities of SOD, CAT, and POD in leaves were observed when exposed to high salinity; which may be associated with diverse metabolic processes. Under diverse stress, enhanced peroxidase activity was associated with cell wall protection from lipid peroxidation, lignification, and cross-linking. APX and GPX are plant peroxidases that sequester chloroplastic H_2_O_2_ by exploiting ascorbate as an electron donor during the first phase of the ascorbate–glutathione cycle [72]. In an NADPH-dependent reaction, GR is required for GSH regeneration in the ascorbate–glutathione chain. In *Macrotyloma uniflorum* [73] and chickpea [74], a higher induction of GR activity was found in tolerant cultivars relative to susceptible variants. A significant interaction effect of MDHAR and DHAR activities implicated notable increases in their activities in leaf tissues compared with roots in the tolerant genotype. Similar results were also reported in maize and wheat [75,76] differing in salt tolerance.

Ascorbate, polyphenols, glutathione, and carotenoid are examples of non-enzymatic antioxidants [23,24]. Glutathione is a non-protein thiol present in plants that serves an important role in protecting them from stressful conditions. At the organ level, GSH levels rose in response to high salt, especially in leaves compared to roots. This might be due to increased sulfur consumption and metabolism for the formation of antioxidants like GSH [77]. Under various conditions, the ascorbate pool may be reduced owing to changes in glutathione levels, which have been linked to ascorbate recycling, or a failure to maintain ascorbate levels, suggesting an overall reduction in strength to endure oxidative stress [75]. Under high salinity, increased sulfate absorption alters GSH levels in tissues [78]. The increased concentration of ASA and GSH, and the lowered redox state of ASA and GSH, suggested that the ASA–GSH cycle is critical for scavenging Free radicals [9]. When exposed to abiotic stressors, the amounts of photosynthetic pigments such as chlorophyll a and b, as well as accessory pigments like carotenoids, reduced as reported by several researchers [79,80,81].

FRAP evaluation measures the capacity of a Fe^3+^/tripyridyl-s-triazine complex to be reduced in the presence of an antioxidant. The formation of non-specific radicals is the function of ABTS and DPPH assay. DPPH activity in the red pericarp sorghum was reported to be higher [82,83,84,85]. Hence, the present results indicated that sorghum grains with colored pericarp had high antioxidant activities. This could be due to the absence of tannins, suggesting that red-colored pericarp sorghum is Type I [83,86] under response to abiotic stresses that lower cytoplasmic osmotic potential [86,87], facilitating water absorption [88].

Plants maintain turgor pressure and osmoregulatory strategies by accumulating osmolytes such polyamines, glycine betaine (GB), proline, and proteins [87,88] under abiotic stresses that lower cytoplasmic osmotic potential, and facilitate water absorption [89]. Proline is the only osmolyte documented to scavenge singlet oxygen and free radicals, particularly hydroxyl ions, and also protects DNA, RNA, and membrane proteins [90]. In the present study, leaves accumulated higher proline levels, particularly at 120 mM NaCl. Glycine betaine (GB) is an osmoregulatory solute that accumulates in plants naturally, especially in leaf tissue [91]. When chloroplasts are exposed to elevated NaCl concentrations, GB can preserve the O_2_ evolving machinery [92]. In sorghum, the salinity-induced synthesis of betaine and BADH mRNA correlates with ABA [93]. The formation of soluble sugars enhances resilience to a variety of stresses. Inside the cell, total soluble carbohydrates serve as osmoprotectants. The exposure of drought or salt stress to sorghum embryos has been shown to result in enhanced sugar levels, which may assist in osmoregulation under stress circumstances [94].

H_2_O_2_ is a naturally occurring toxic plant cellular metabolite that induces thylakoid degradation [95,96]. Enhanced antioxidant activities, proline, and total carbohydrate concentration in sensitive genotypes may be linked to higher H_2_O_2_ formation. H_2_O_2_ concentration was approximately 52% higher in salt-stressed plants than control plants [97]. Under stressful situations, the membrane injury index in terms of electrolyte leakage increased significantly and was highest in root tissues. Accelerated electrolyte leakage within tissues is probably the result of altered cellular membrane physical properties. Cell membrane damage is caused by malondialdehyde (MDA), a result of the peroxidation of unsaturated fatty acids in phospholipids. The results of this study found that salt stress enhanced MDA formation, which is consistent with previous studies [98,99]. In wild-type and transgenic sorghum lines, salt stress-induced cell membrane peroxidation causes membrane permeability, which results in increased electrolytic leakage [100]. During stress conditions, the total polyamine concentration was increased. The elevated content of several polyamines in sorghum leaf tissues showed that they may play a function in ROS scavenging [101,102,103].

The analysis of yield components is an essential tool for assessing crop development and productivity [104]. Reduced yield attributes were elevated at higher salinity levels because of the adverse effect on the physiological process and dry matter accumulation and, ultimately, reduced the plant’s seed yield. The reduction in the number of seeds per plant was due to reduced flower production and fertilization. Seed yield was affected due to pollen sterility, abortion, pollen germination, and in-compatible fertilization, which directly reduced yield attributes and yield under salt stress [105].

The differential expression of ion transporter genes under different salinity levels indicated their adaptive behavior for Na^+^ ion exclusion. *NHX* proteins are ubiquitous transmembrane specialized proteins that maintain ion homeostasis by sequestering excess Na^+^ ions in vacuoles or eliminating them from the cells [18]. In our study, the expression of genes encoding the Na^+^/H^+^ and H(^+^)- exporting diphosphatase exchangers, which are crucial for the transport of Na^+^/K^+^ and the maintenance of ion homeostasis, were upregulated in the tolerant genotype. Salt stress elicits a cytosolic calcium signal [106]. The increased intracellular concentration of Ca^2+^ ions is sensed by *SOS3*, which interacts and activates *SOS2*, a serine/threonine-protein kinase (Figure 13). Both *SOS2* and *SOS3* regulate the expression level of *SOS1*, a salt tolerance gene that encodes a sodium proton antiporter (*NHX*) [107]. The co-expression of *SOS1* and *SOS2* together with *NHX-1* dramatically enhances the salt tolerance capacity in plants. In isolated plasma membrane vesicles isolated from wild-type plants, a constitutively active *SOS2* kinase boosts an *NHX* exchange activity. The plasma membrane *NHX* exchange activity is considerably reduced in sos2-2 and sos3-1 mutants. Activated *SOS2* can be added to membrane vesicle preparations in vitro to restore it to near-wild-type levels [108,109].

## 5. Conclusions

This study has shown that at different developmental stages, salinity had a profound effect on the ability sorghum tissues to cope with the associated oxidative stress from the arsenal of alternate antioxidant mechanisms. Moreover, several phenotypic and biochemical traits exhibited differential behavior to high salinity. At early stages, both leaves and roots developed more efficient metabolic mechanisms specific to different organs, while at later stages, phenotypic traits and antioxidant scavengers contributed the most to the differentiation among stressed and non-stressed plants. Additionally, this work gives valuable information and insight into the phenotypic and physio-biochemical behaviors of contrasting sorghum genotypes, making it possible to decipher the genetics underpinning salt tolerance. These findings potentially pave the way for improving salt tolerance by exploiting suitable candidate genes and associated regulatory networks with the development of effective genome-assisted breeding strategies for the genetic improvement of sorghum.

## Figures and Tables

**Figure 1 ijms-22-13249-f001:**
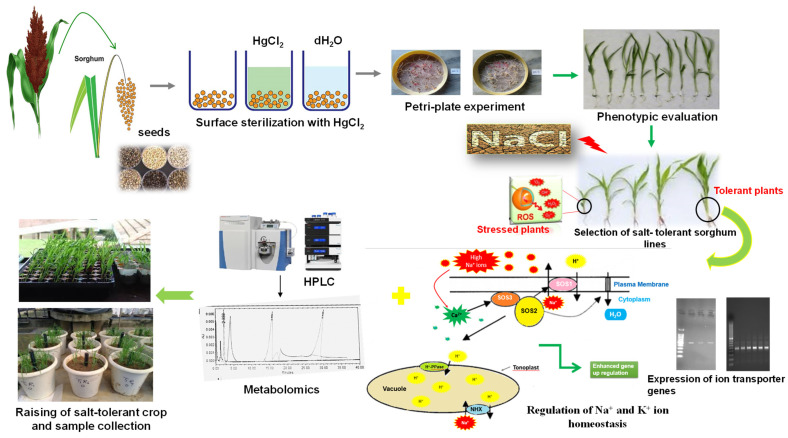
Outline of the experiment/Two *S. bicolor* genotypes (SSG 59-3 and PC-5) with varied salt tolerance were phenotypically evaluated based on germination studies and then planted in screen house under different salt concentrations. Leaves and roots from five plants were pooled and considered a biological replicate, and two such replicates were used in the analysis. HPLC was performed to evaluate the accumulation of polyamines under different salt concentrations. Total RNA was isolated from sorghum leaves and the expression pattern of ion transporters was studied.

**Figure 2 ijms-22-13249-f002:**
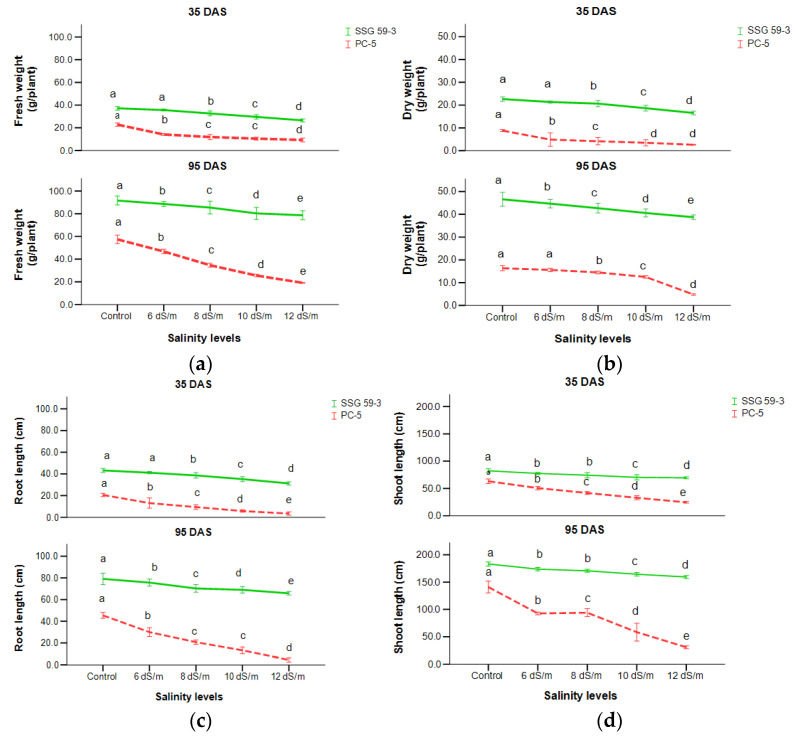
Effect of salt stress on (**a**) fresh weight, (**b**) dry weight, (**c**) root length, and (**d**) shoot length of sorghum genotypes at 35 and 95 DAS. Post hoc comparisons of the means were performed using Tukey’s HSD test at *p* < 0.05; different letters (^a–e^) indicate significant differences among treatments within each genotype.

**Figure 3 ijms-22-13249-f003:**
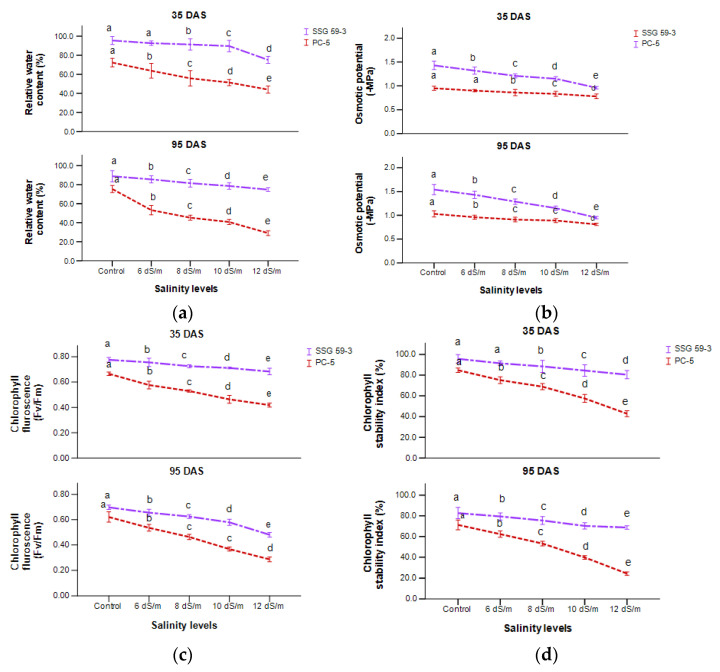
Effect of salt stress on relative water content, (**a**) osmotic potential, (**b**) chlorophyll fluorescence, and (**c**) chlorophyll stability index (**d**) of sorghum genotypes at 35 and 95 DAS. Post hoc comparisons of the means were performed using Tukey’s HSD test at *p* < 0.05; different letters (^a–e^) indicate significant differences among treatments within each genotype.

**Figure 4 ijms-22-13249-f004:**
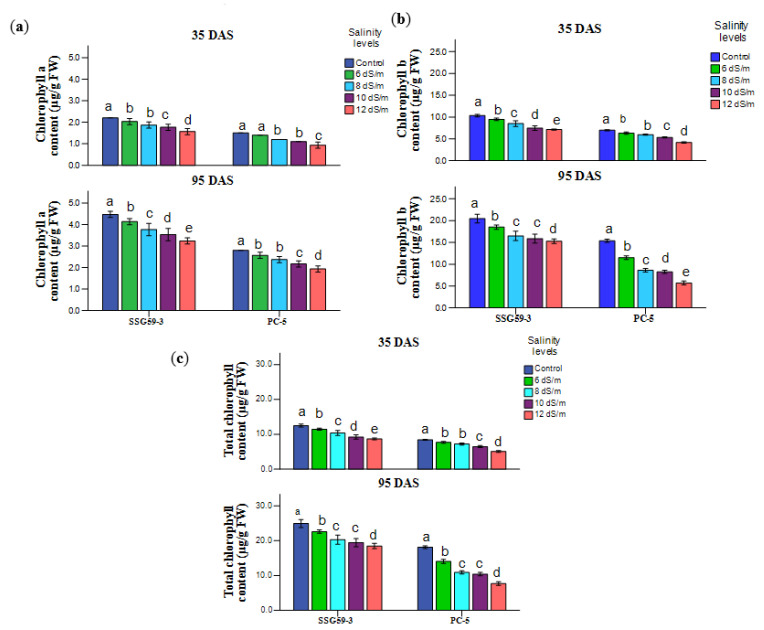
Effect of salt stress on (**a**) chlorophyll a, (**b**) chlorophyll b, and (**c**) total chlorophyll of sorghum genotypes at 35 and 95 DAS. Post hoc comparisons of the means were performed using Tukey’s HSD test at *p* < 0.05; different letters (^a–e^) indicate significant differences among treatments within each genotype.

**Figure 5 ijms-22-13249-f005:**
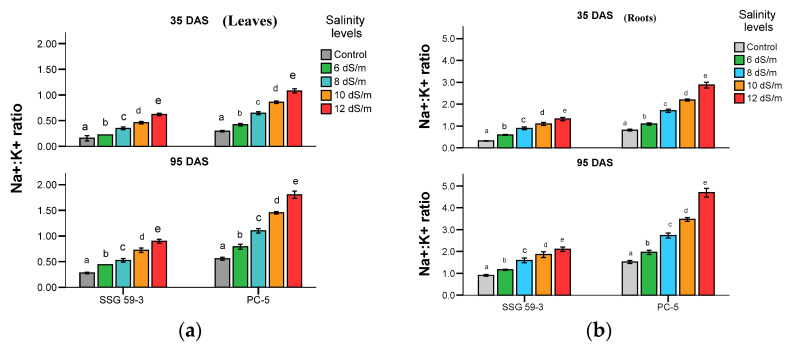
Effect of salt stress on Na^+^:K^+^ in leaves (**a**), and Na^+^:K^+^ in roots (**b**) of sorghum genotypes at 35 and 95 DAS. Post hoc comparisons of the means were performed using Tukey’s HSD test at *p* < 0.05; different letters (^a–e^) indicate significant differences among treatments within each genotype.

**Figure 6 ijms-22-13249-f006:**
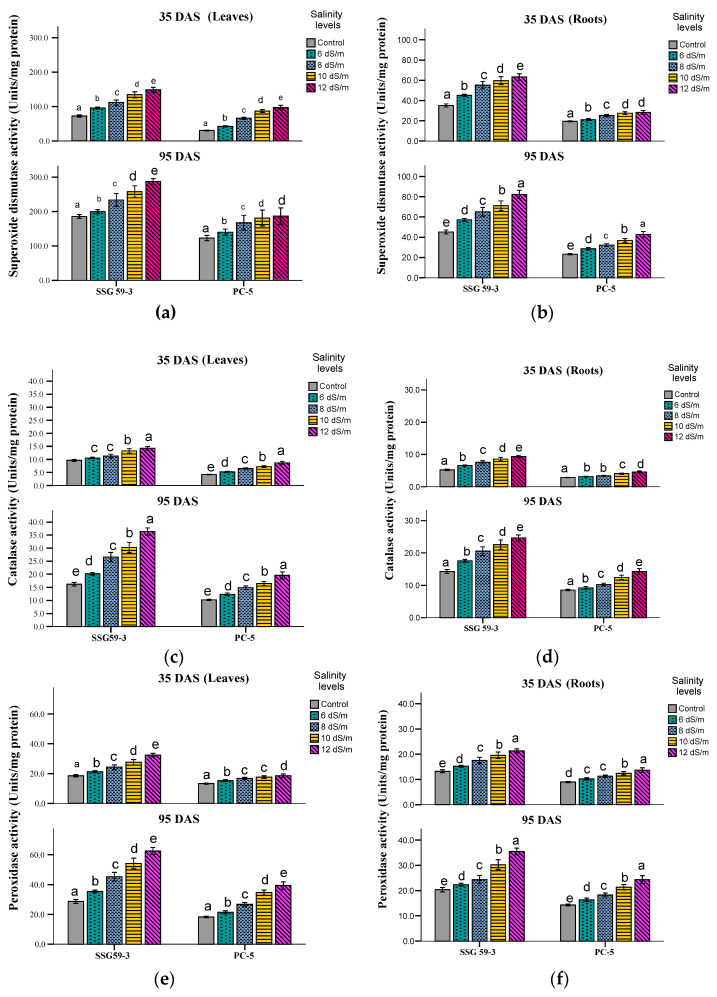
Effect of salt stress on superoxide dismutase (SOD, (**a**,**b**)), catalase (CAT, (**c**,**d**)), peroxidase (POX, (**e**,**f**)), and ascorbate peroxidase (APX, (**g**,**h**)) of sorghum genotypes at 35 and 95 DAS. Post hoc comparisons of the means were performed using Tukey’s HSD test at *p* < 0.05; different letters (^a–e^) indicate significant differences among treatments within each genotype.

**Figure 7 ijms-22-13249-f007:**
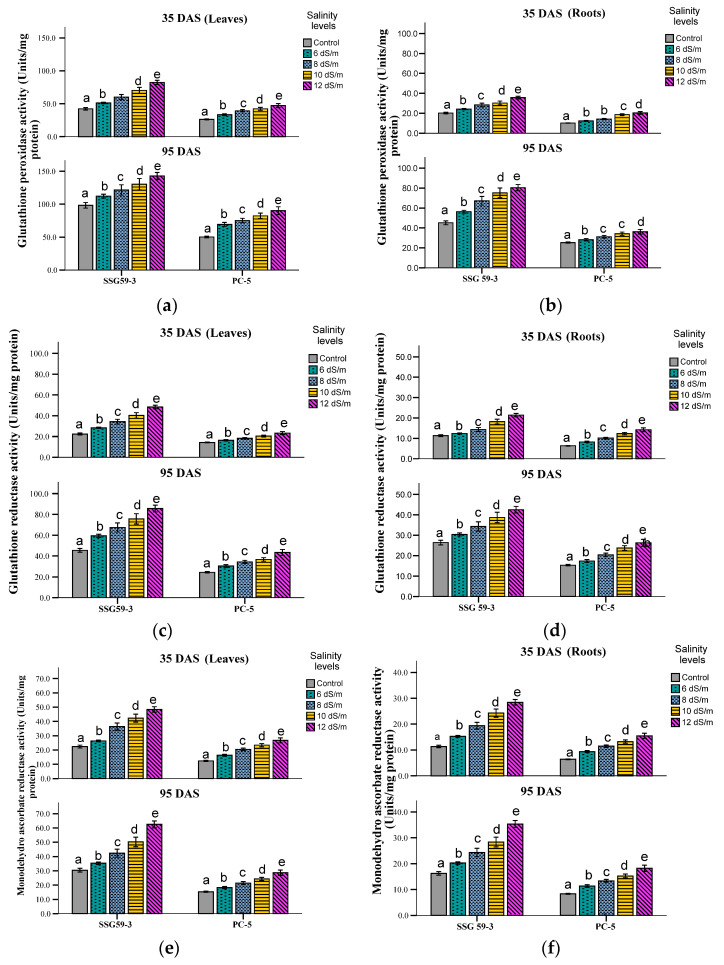
Effect of salt stress on glutathione peroxidase (GPX, (**a**,**b**)), glutathione reductase (GR, (**c**,**d**)), monodehydroascorbate reductase (MDHAR, (**e**,**f**)), and dehydroascorbate reductase (DHAR, (**g**,**h**)) of sorghum genotypes at 35 and 95 DAS. Post hoc comparisons of the means were performed using Tukey’s HSD test at *p* < 0.05; different letters (^a–e^) indicate significant differences among treatments within each genotype.

**Figure 8 ijms-22-13249-f008:**
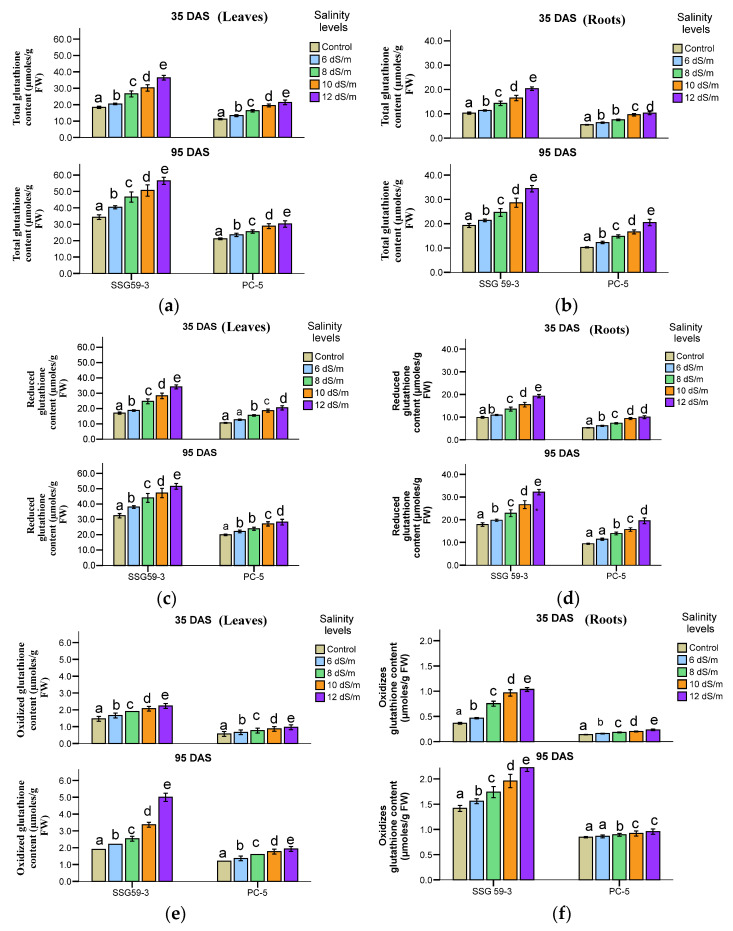
Effect of salt stress on total glutathione (**a**,**b**), reduced glutathione (GSH, (**c**,**d**), and oxidized glutathione (GSSG, (**e**,**f**)) of sorghum genotypes at 35 and 95 DAS. Post hoc comparisons of the means were performed using Tukey’s HSD test at *p* < 0.05; different letters (^a–e^) indicate significant differences among treatments within each genotype.

**Figure 9 ijms-22-13249-f009:**
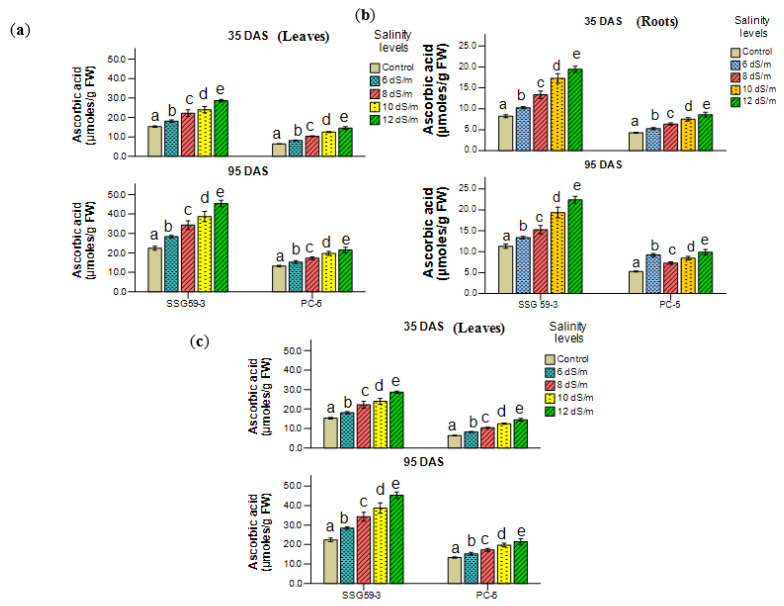
Effect of salt stress on (**a**,**b**) ascorbic acid and (**c**) carotenoids of sorghum genotypes at 35 DAS and 95 DAS. Post hoc comparisons of the means were performed using Tukey’s HSD test at *p* < 0.05; different letters (^a–e^) indicate significant differences among treatments within each genotype.

**Figure 10 ijms-22-13249-f010:**
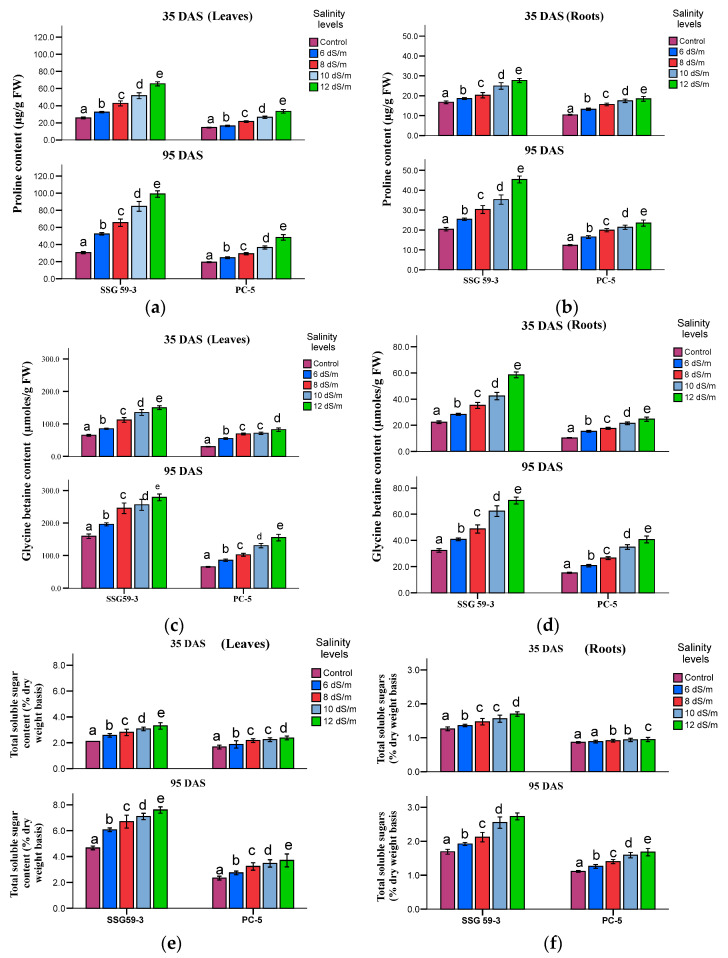
Effect of salt stress on proline (Pro, (**a**,**b**)), glycine betaine (GB, (**c**,**d**)), and total soluble carbohydrates (TSC, (**e**,**f**)) of sorghum genotypes at 35 and 95 DAS. Post hoc comparisons of the means were performed using Tukey’s HSD test at *p* < 0.05; different letters (^a–e^) indicate significant differences among treatments within each genotype.

**Figure 11 ijms-22-13249-f011:**
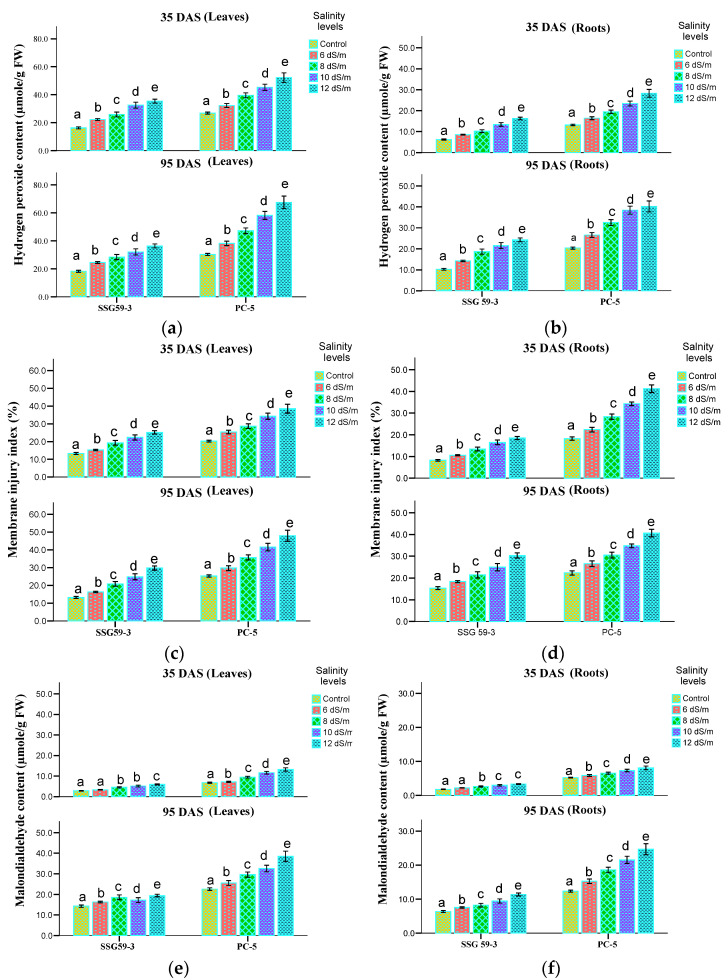
Effect of salt stress on hydrogen peroxide (H_2_O_2_, (**a**,**b**)), membrane injury index (MII, (**c**,**d**)), and malondialdehyde (MDA, (**e**,**f**)) of sorghum genotypes at 35 and 95 DAS. Post hoc comparisons of the means were performed using Tukey’s HSD test at *p* < 0.05; different letters (^a–e^) indicate significant differences among treatments within each genotype.

**Figure 12 ijms-22-13249-f012:**
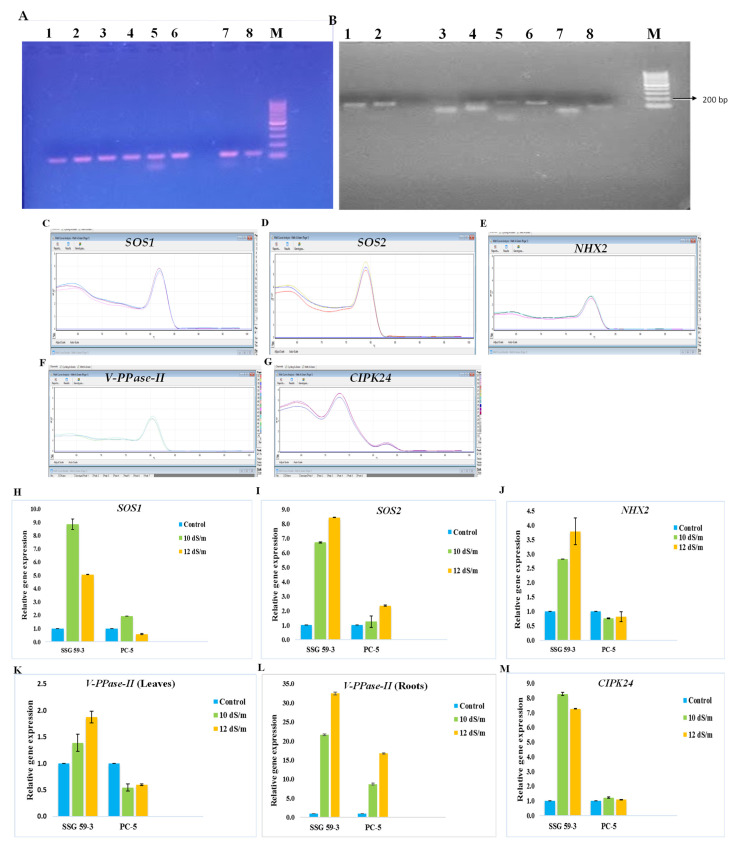
Agarose gel: (**A**) mRNA of salt stress-responsive genes (1: *SOS1*; 2: *SOS2*; 3. *NHX2*; 4: *V-PPase-II* (leaves); 5: *V-PPase-II* (roots); 6: *CIPK24*; 7: *Act1*; 8: *PP2A*; L: ladder)), (**B**) PCR products of the expected sizes (≤ 200 bp; 1: *SOS1*; 2: *SOS2*; 3. *NHX2*; 4: *V-PPase-II* (leaves); 5: *V-PPase-II* (roots); 6: *CIPK24*; 7: *Act1*; 8: *PP2A*; L: ladder)), (**C**–**G**) melt/dissociation curve of salt stress-responsive genes under saline conditions, (**H**–**M**) the expression levels of stress-related ion transporter genes based on qPCR. The *PP2A* gene was used as the reference gene/internal control. Data are shown as mean ± S.D. (*n* = 3).

**Figure 13 ijms-22-13249-f013:**
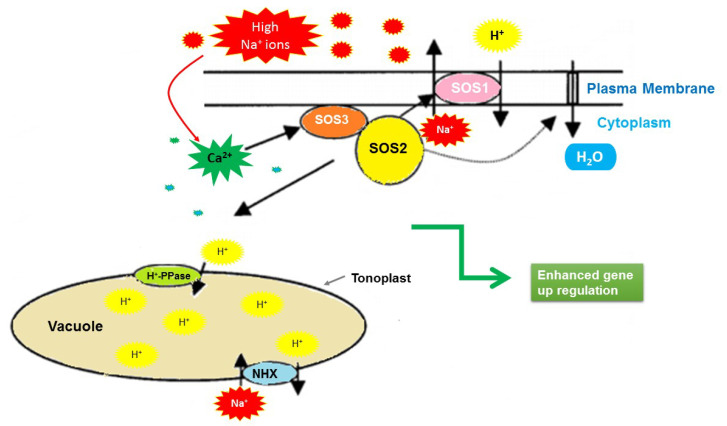
Regulation of Na^+^ and K^+^ ion homeostasis by the SOS pathway. Excess Na^+^ ions elicit a calcium signaling pathway that activates the *SOS1–SOS3* protein kinase complex, which stimulates the *SOS1* and sodium proton antiporter (*NHX-2*) exchange activity and, thus, regulates the expression of genes encoding them.

**Table 1 ijms-22-13249-t001:** List of primers.

Gene	Accession No.	Primers	Sequence (5′-3′)
*SOS1*	XM_015763865.2	Forward	GGCTCAAGGCCAACTGGTAT
		Reverse	TTTCGGGCCTCCCTATCTCA
*SOS2*	KP330207.1	Forward	TGCATCAGTACTGTGGCTGG
		Reverse	ATTTAGAAGCCGCACACGGA
*NHX-2*	EU482408.2	Forward	CGATGGGTGAACGAGTCCAT
		Reverse	GTTGCAAAAGTATGTCTGGCA
*V-PPase-11*	GQ469975.1	Forward	GCTACGGCGACTACCTCATC
		Reverse	CCTTCGGAGATAGCGTTCTG
*CIPK24*	XM_002438609.2	Forward	TCTCCAGGAGCCAAGTCATT
		Reverse	CAAACCATGGGTCTGCTCTT
*PP2A*	XM_002448914.2	Forward	AAAAGGCTGCAGAAACGAAG
		Reverse	GCTTCAATTGGGGCAGATAA

**Table 2 ijms-22-13249-t002:** Antioxidant activity and polyphenolic composition of sorghum grains.

Genotypes	DPPH (mg TE/100 g)	FRAP mg (TE/100 g)	ABTS (mg/100 g)	TPC (mg GAE /100 g)	TFC (mg RE/100 g)	Flavanols (mg/100 g DW)	o-HP (mg/100 g DW)
**SSG 59-3**	22.4 ± 0.11 ^a^	16.4 ± 0.1 ^a^	43.7 ± 0.23 ^a^	940.7 ± 6.7 ^a^	45.3 ± 0.35 ^a^	0.292 ± 0.052 ^a^	0.924 ± 0.21 ^a^
**PC-5**	9.7 ± 0.07 ^b^	7.1 ± 0.25 ^b^	20.2 ± 0.18 ^b^	291.2 ± 3.8 ^b^	17.2 ± 0.11 ^b^	0.117 ± 0.23 ^b^	0.243 ± 0.05 ^b^

^a–e^ Values with different superscripts in the same row are significantly different at *p* < 0.05. DPPH (2, 2-diphenyl-1-picrylhydrazyl), FRAP (Ferric reducing the ability of plasma assay), ABTS (2,4,6-Tris(2-pyridyl)-s-triazine), and TE (Trolox equivalents), DW (Dry weight basis), GAE (Gallic acid equivalents), RE (Rutin equivalents), TPC (Total phenolic content), TFC (Total flavonoid content), and o-HP (o-Hydoxyphenols).

**Table 3 ijms-22-13249-t003:** Changes in polyamines (Put, Spd, and Spm) in the flag leaf of sorghum genotypes under salt stress.

		35 DAS			95 DAS		
Genotypes	Treatments	Put *	Spd *	Spm *	Put	Spd	Spm
**SSG 59-3**	Control	270.9 ± 1.2 ^a^	246.8 ± 1.1 ^e^	15.5 ± 0.98 ^a^	214.6 ± 1.3 ^a^	195.3 ± 0.96 ^e^	11.1 ± 0.12 ^a^
	60 mM	336.5 ± 1.4 ^b^	271.2 ± 1.2 ^d^	19.5 ± 0.65 ^b^	265.9 ± 1.2 ^b^	228.2 ± 1.3 ^d^	14.5 ± 0.16 ^b^
	80 mM	382.5 ± 1.8 ^c^	287.5 ± 1.4 ^c^	25.6 ± 0.62 ^c^	319.6 ± 1.1 ^c^	245.8 ± 1.4 ^c^	17.6 ± 0.19 ^c^
	100 mM	457.6 ± 2.3 ^d^	314.2 ± 1.9 ^b^	39.6 ± 0.34 ^d^	384.7 ± 1.9 ^d^	294.7 ± 1.5 ^b^	20.6 ± 0.21 ^d^
	120 mM	496.4 ± 2.8 ^e^	357.9 ± 2.0 ^a^	45.2 ± 0.64 ^e^	402.6 ± 2.5 ^e^	320.1 ± 2.3 ^a^	23.9 ± 0.25 ^e^
**PC-5**	Control	155.7 ± 1.3 ^a^	123.6 ± 0.68 ^e^	6.4 ± 0.23 ^a^	98.6 ± 0.26 ^a^	88.6 ± 0.67 ^e^	4.6 ± 0.03 ^a^
	60 mM	191.8 ± 1.2 ^b^	134.2 ± 1.1 ^d^	10.8 ± 0.37 ^b^	110.6 ± 1.3 ^b^	99.3 ± 0.95 ^d^	5.7 ± 0.12 ^b^
	80 mM	233.5 ± 1.4 ^c^	141.5 ± 1.2 ^c^	14.6 ± 0.56 ^c^	138.6 ± 0.98 ^c^	117.3 ± 1.2 ^c^	8.5 ± 0.06 ^c^
	100 mM	297.7 ± 1.5 ^d^	164.3 ± 1.3 ^b^	18.6 ± 0.61 ^d^	180.2 ± 1.2 ^d^	126.8 ± 1.1 ^b^	10.5 ± 0.16 ^d^
	120 mM	324.1 ± 1.7 ^e^	189.4 ± 1.4 ^a^	26.6 ± 0.62 ^e^	213.6 ± 1.4 ^e^	140.3 ± 1.6 ^a^	12.7 ± 0.20 ^e^

* nmol g^−1^ FW; Put: Putrescine; Spd: spermidine; Spm: spermine. ^a–e^ Values with different superscripts in the same row are significantly different at *p* < 0.05; DAS: days after sowing.

**Table 4 ijms-22-13249-t004:** Yield attributing traits in sorghum under salt stress.

Stages	35 DAS			95 DAS				
Genotypes	SSG 59-3	PC-5	SSG 59-3	PC-5	SSG 59-3	PC-5	SSG 59-3	PC-5
**Character**	Plant Height		Number of Leaves/Plant	Plant Height		Number of Leaves/Plant		
**Control**	77.3 ± 1.2 ^a^	67.7 ± 2.1 ^a^	10.1 ± 0.23 ^a^	6 ± 0.06 ^a^	190.1 ± 22.6 ^a^	154.2 ± 11.3 ^a^	13 ± 0.06 ^a^	8 ± 0.11 ^a^
**60 mM**	72.3 ± 2.3 ^b^	53.1 ± 1.1 ^b^	9.1 ± 0.45 ^b^	5 ± 0.11 ^b^	176.5 ± 15.6 ^b^	123.4 ± 13.2 ^b^	10 ± 0.11 ^b^	7 ± 0.10 ^b^
**80 mM**	67.8 ± 2.4 ^c^	41.6 ± 1.5 ^c^	8.2 ± 0.12 ^c^	4 ± 0.14 ^c^	158.6 ± 14.2 ^c^	81.4 ± 13.5 ^c^	9 ± 0.06 ^c^	6 ± 0.13 ^c^
**100 mM**	57.7 ± 1.6 ^d^	34.5 ± 1.9 ^d^	7.3 ± 0.16 ^d^	4 ± 0.16 ^c^	145.1 ± 13.3 ^d^	64.4 ± 14.6 ^d^	8 ± 0.07 ^d^	6 ± 0.06 ^c^
**120 mM**	49.2 ± 1.9 ^e^	24.6 ± 1.4 ^e^	6.1 ± 0.21 ^e^	3 ± 0.09 ^d^	128.1 ± 12.6 ^e^	53.6 ± 12.8 ^e^	7 ± 0.09 ^e^	4 ± 0.09 ^d^

^a–e^ Values with different superscripts in the same row are significantly different at *p* < 0.05; DAS: days after sowing.

**Table 5 ijms-22-13249-t005:** Seed yield and number of tiller traits in sorghum at physiological maturity.

	Physiological Maturity (95 DAS)			
Genotypes	SSG 59-3	PC-5	SSG 59-3	PC-5
**Character**	Number of Tiller/Plant		Seed Yield/Plant (g)	
**Control**	5.03 ± 0.10 ^a^	3 ± 0.13 ^a^	15.25 ± 0.13 ^a^	9.56 ± 0.11 ^a^
**60 mM**	4 ± 0.09 ^b^	2 ± 0.06 ^b^	13.26 ± 0.18 ^b^	7.35 ± 0.09 ^b^
**80 mM**	3 ± 0.04 ^c^	1.73 ± 0.1 ^c^	12.43 ± 0.16 ^c^	6.12 ± 0.04 ^c^
**100 mM**	2.6 ± 0.05 ^d^	1.33 ± 0.09 ^d^	11.32 ± 0.14 ^d^	5.48 ± 0.16 ^d^
**120 mM**	2 ± 0.03 ^d^	0.95 ± 1.2 ^e^	9.57 ± 0.16 ^e^	3.26 ± 0.05 ^e^

^a–e^ Values with different superscripts in the same row are significantly different at *p* < 0.05; DAS: days after sowing.

## Data Availability

Data are contained within the article or Appendix A.

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
