# Peer review of "Ascorbate–Glutathione Oxidant Scavengers, Metabolome Analysis and Adaptation Mechanisms of Ion Exclusion in Sorghum under Salt Stress"

_ijms, 2021, doi:10.3390/ijms222413249_

Round 1
Reviewer 1 Report
The authors have demonstrated that two genotypes of sorghum displayed different salt responses. Please improve the presentation of the figures according to the following suggestions.
- How many plants are included in one replicate?
- Please use one type of statistical analysis. Use one label (either letters or asterisks).
- In Figure 2, 3, and 4, are the statistical analyses done between genotypes or among different treatments? Please make the label on the figures.
- In Figure 5 to 7, the size of the fonts is not equal.
Does the same letter mean no difference in one figure? For example, in Figure 5, 10 ds/m in both genotypes showed NO difference? Use one type of label. Both * (asterisk) and letter are confusing. - In Figure 8 to 10, Asterisk is compared with the control of each genotype?
- Figure 11, where are the statistical labels?
- Figure 12, please repeat this experiment with real-time RT-PCR. The level of actin in (e) was saturated, so it's not suitable to be the internal control.
- In Table 2, should be a-e. And the comparisons should be made with one metabolite in two genotypes, right? So, should compare the values in the same "column", not "row".
- In Table 3, add (35 DAS) and (95DAS) in the first row, respectively. How to read the statistical labels (a to e)?
- In Table 4, add (35 DAS) and (95DAS) in the first row, respectively. What do a and b mean in the description?
- In Line 51, "photosynthesis," might need to be removed.
- In Line 58 to 60, the sentence "It had wider adaptability under..." seemed grammar incorrect.
- In Line 66, "S. bicolor" should be Italic.
- In Figure 1, the photo of "Metabolomics" is not well aligned. May need to adjust.
Author Response
Pointwise answers to reviewer’s comments
English language and style are needful done as highlighted and minor spell check is also needful done.
|
S.No |
Comment |
Reply |
|
|
Reviewer #1: |
|
|
|
Are the methods adequately described? Can be improved |
The methods have been improved as suggested.
Kindly refer line no. 111-361. |
|
|
Are the results clearly presented? Can be improved |
The results have been improved as suggested.
Kindly refer line no. 362-1349. |
|
|
Are the conclusions supported by the results? Can be improved |
Rewrote the conclusion to support the findings of the present study.
Kindly refer line no. 1523-1536. |
|
|
Please improve the presentation of the figures according to the following suggestions: |
|
|
1 |
How many plants are included in one replicate? |
Five plants per treatment were included in one replicate.
Kindly refer line no. 127. |
|
2 |
Please use one type of statistical analysis. Use one label (either letters or asterisks). |
Letters have been used for statistical analysis in all Figures (Fig. 2 to 11).
Kindly refer line no. 464-475 (Figure 2), 475-481 (Figure 3) and 518-524 (Figure 4), 542-546 (Figure 5), 630-645 (Figure 6) and 664-680 (Figure 7), 721-728 (Figure 8), 756-761 (Figure 9) and 841-848 (Figure 10), 878-895 (Figure 11). |
|
3 |
In Figures 2, 3, and 4, are the statistical analyses done between genotypes or among different treatments? Please make the label on the figures |
The labels have been added in Figures 2, 3 and 4 as suggested.
Kindly refer line no. 464-475 (Figure 2), 475-481 (Figure 3) and 518-524 (Figure 4). |
|
4 |
In Figures 5 to 7, the size of the fonts is not equal. Does the same letter mean no difference in one figure? For example, in Figure 5, 10 ds/m in both genotypes showed NO difference? Use one type of label. Both * (asterisk) and letter are confusing. |
Font size is adjusted. The labels have been used in Figures 5 to 7. * (asterisk) were removed.
Kindly refer line no. 542-546 (Figure 5), 630-645 (Figure 6) and 664-680 (Figure 7). |
|
5 |
In Figure 8 to 10, Asterisk is compared with the control of each genotype? |
The labels have been added in Figures 8 to 10 as suggested. The comparisons have been made concerning control plants (non-stressed).
Kindly refer line no. 721-728 (Figure 8), 756-761 (Figure 9) and 841-848 (Figure 10) |
|
6 |
Figure 11, where are the statistical labels? |
The statistical labels have been added in Figure 11.
Kindly refer line no. 878-895. |
|
7 |
Figure 12, please repeat this experiment with real-time RT-PCR. The level of actin in (e) was saturated, so it's not suitable to be the internal control. |
Real-time RT-PCR (qPCR) has been performed. The data is presented in Figure 12.
Kindly refer line no. 1129-1143. |
|
8 |
Table 2, should be a-e. And the comparisons should be made with one metabolite in two genotypes, right? So, should compare the values in the same "column", not "row". |
The comparisons have been made with one metabolite in two genotypes as suggested in Table 2.
Kindly refer line no. 762-767. |
|
9 |
In Table 3, add (35 DAS) and (95DAS) in the first row, respectively. How to read the statistical labels (a to e)? |
Added (35 DAS) and (95 DAS) in the first row in Table 3. The statistical labels (a to e) can be read from control to 120 mM NaCl in the column for 1 metabolite, similar to Table 2.
Kindly refer line no. 908-911. |
|
10 |
In Table 4, add (35 DAS) and (95DAS) in the first row, respectively. What do a and b mean in the description? |
Added (35 DAS) and (95 DAS) in the first row in Table 4. a and b were typographic errors.
Kindly refer line no. 1171-1173. |
|
11 |
In Line 51, "photosynthesis," might need to be removed. |
Removed "photosynthesis".
Kindly refer no. 63. |
|
12 |
In Line 58 to 60, the sentence "It had wider adaptability under..." seemed grammar incorrect. |
The line has been rewritten as suggested.
Kindly refer no. 70-71. |
|
13 |
In Line 66, "S. bicolor" should be Italic. |
"S. bicolor" has been written in Italics.
Kindly refer no. 78. |
|
14 |
In Figure 1, the photo of "Metabolomics" is not well aligned. May need to adjust. |
Aligned the photo of "Metabolomics" in Figure 1.
Kindly refer no. 77-88. |

Reviewer 2 Report
The manuscript present data on sorghum plants response to salinity stress using two cultivars of contrasting tolerance.
The impact of the stress on plants has been widely reported in literature. The results based on several physiological, biochemical and molecular methods obtained by the authors are generally consistent with those obtained for other species.
The novelty lies mostly in that the authors compared young and old plant organs. But it shuld be emphasised more visibly.
The beneficial side of the researches is use of multiplicity of methods that provides quite deep and widespread description of response.
The weakest side of the manuscript is statistics. It should be considerably improved. The authors declare in the Methods use of ANOVA and Tukey test. The results of the analysis are not presented properly in the text.
Papers given, for example, below:
Lazarević, B., Šatović, Z., Nimac, A., Vidak, M., Gunjača, J., Politeo, O. and Carović-Stanko, K., 2021. Application of phenotyping methods in detection of drought and salinity stress in Basil (Ocimum basilicum L.). Frontiers in Plant Science, 12, p.174.
Nguyen, H.M., Sako, K., Matsui, A., Suzuki, Y., Mostofa, M.G., Ha, C.V., Tanaka, M., Tran, L.S.P., Habu, Y. and Seki, M., 2017. Ethanol enhances high-salinity stress tolerance by detoxifying reactive oxygen species in Arabidopsis thaliana and rice. Frontiers in Plant Science, 8, p.1001.
Javadipour, Z., Balouchi, H., Movahhedi Dehnavi, M. et al. Physiological Responses of Bread Wheat (Triticum aestivum) Cultivars to Drought Stress and Exogenous Methyl Jasmonate. J Plant Growth Regul (2021). https://doi.org/10.1007/s00344-021-10525-w
and many others published in high quality journals could be helpful.
There are several mistakes, cases of incorrect/ unclear use of scientific terms:
For example:
line 26 .. "photosynthetic rate" - was it measured?
line 139 .. "Root and shoot length was calculated" .... rather measured ..
line 143 .... "temperature in diffused light for six h." ... rather hours
line 336 and other ..Values are ..... significant differences between means -- it is not clear for me
line 287 and other Tuckey's test - should be Tukey or Tukey's test
line 294 and other ...among both genotypes at p<0.05... sholud be (P<0.05)
line 321 ... - What is meant by "the reduction in photosynthetic assembly"
line 370 and other "* and ** represent the most significant and highly
significant differences among the genotypes and salt treatment"
- not precise, P-values should be given
line 628 ...."Statistical difference was calculated by standard deviation" - std dev does nor serve for testing statistical differences.
Figure 4 - lacks of Tukey comparisons
Table 4 needs improvement
190.10±22.6 badly written. the same decimals should be used 190.1±22.6
also in remained cases
Which values are given in tables 4 and 5? Means and SD ?
Discussion needs improvement to highlight own results and discuss them with those reported in literature.
The Conclusions are not adequate to obtained results.
Author Response
Pointwise answers to reviewer’s comments
|
S.No |
Comment |
Reply |
|
|
Reviewer #2: |
|
|
|
Moderate English changes required |
Moderate English language and style are needful done as highlighted and minor spell check is also needful done. |
|
|
Are the results clearly presented? Can be improved |
The results have been improved as suggested.
Kindly refer line no. 362-1349. |
|
|
Are the conclusions supported by the results? Must be improved |
Rewrote the conclusion to support the findings of the present study.
Kindly refer line no. 1522-1536. |
|
|
The novelty lies mostly in that the authors compared young and old plant organs. But it should be emphasized more visibly. |
The results have been emphasized young and old plant organs.
Kindly refer line no. 361-1348. |
|
|
The weakest side of the manuscript is statistics. It should be considerably improved. The authors declare in the Methods use of ANOVA and Tukey test. The results of the analysis are not presented properly in the text. |
The figures have been modified and the statistical labels have been added in all Figures. |
|
|
Papers are given, for example, below: 2. Nguyen, H.M., Sako, K., Matsui, A., Suzuki, Y., Mostofa, M.G., Ha, C.V., Tanaka, M., Tran, L.S.P., Habu, Y. and Seki, M., 2017. Ethanol enhances high-salinity stress tolerance by detoxifying reactive oxygen species in Arabidopsis thaliana and rice. Frontiers in Plant Science, 8, p.1001. 3. Javadipour, Z., Balouchi, H., Movahhedi Dehnavi, M. et al. Physiological Responses of Bread Wheat (Triticum aestivum) Cultivars to Drought Stress and Exogenous Methyl Jasmonate. J Plant Growth Regul (2021). https://doi.org/10.1007/s00344-021-10525-w and many others published in high-quality journals could be helpful. |
The papers have been used as a reference to improve the manuscript. |
|
|
||
|
|
There are several mistakes, cases of incorrect/ unclear use of scientific terms: |
|
|
1 |
line 26 .. "photosynthetic rate" - was it measured? |
Deleted the word rate.
Kindly refer line no. 35 |
|
2 |
line 139 .. "Root and shoot length was calculated" .... rather measured. |
Removed the word “calculated” and replaced it with “measured”.
Kindly refer line no. 171. |
|
3 |
line 143 .... "temperature in diffused light for six h." ... rather hours |
Removed the word “h” and replaced it with “hours”.
Kindly refer line no. 176. |
|
4 |
line 336 and other ..Values are ..... significant differences between means -- it is not clear for me |
Each dot in the line graphs represents the means of three biological replicates. The line has been rewritten to make it clear and precise.
Kindly refer line no. 473-475. |
|
5 |
line 287 and other Tuckey's test - should be Tukey or Tukey's test |
Rectified the word as “Tukey's” and at other places.
Kindly refer line no. 359. |
|
6 |
line 294 and other ...among both genotypes at p<0.05... should be (P<0.05) |
Replaced the word with P<0.05 and at other places.
Kindly refer line no. 359, 367, 474, 480, 523, 545, 644, 679, 727, 760, 763, 776, 847, 869, 894, 911. |
|
7 |
line 321 ... - What is meant by "the reduction in photosynthetic assembly" |
Deleted the word “photosynthetic assembly”.
Kindly refer line no. 454. |
|
8 |
line 370 and other "* and ** represent the most significant and highly significant differences among the genotypes and salt treatment" |
* (asterisk) were removed and replaced with statistical labels. The comparisons have been made concerning control plants (non-stressed).
Kindly refer line no. 544-546. |
|
9 |
- not precise, P-values should be given |
Needful done. |
|
10 |
line 628 ...."Statistical difference was calculated by standard deviation" - std dev does nor serve for testing statistical differences. |
Removed the line...."Statistical difference was calculated by standard deviation" from the text as suggested.
Kindly refer line no. 1142-1143. |
|
11 |
Figure 4 - lacks Tukey comparisons |
The statistical labels have been added in Figure 4.
Kindly refer line no. 518-524. |
|
12 |
Table 4 needs improvement |
Table 4 has been improved and statistical labels have been added for comparison.
Kindly refer line no. 1171-1173. |
|
13 |
190.10±22.6 badly written. the same decimals should be used 190.1±22.6 |
Rectified the decimal errors and at other places.
Kindly refer line no. 1171-1173. |
|
14 |
Which values are given in tables 4 and 5? Means and SD ? |
The mean values of three replicates and standard error. |
|
15 |
The discussion needs improvement to highlight own results and discuss them with those reported in the literature. |
The discussion has been improved as suggested.
Kindly refer line no. 1350-1515. |
|
16 |
The Conclusions are not adequate to obtain the results. |
Rewrote the conclusion to support the findings of the present study.
Kindly refer line no. 1523-1536. |

Round 2
Reviewer 2 Report
The manuscript concerns the plant response to salinity that is analysed through comparison of two sorghum genotypes of contrasting tolerance. The studies were performed on the basis of experiments at a controlled environment using multiple techniques, including biochemical, physiological and molecular ones that gives quite deep insight into the problem. The main novelty lies in that the response is compared at different plant developmental stages.
This version shows substantial improvent, and obvious mistakes have been removed. In my opinion this manuscript can a valuable contribution in the area of salinity stress, that has a worldwide detrimental impact on agronomic plant production and environmental systems.
Some notes:
Probably text format should be corrected at:
71. BHATTACHARJEE, S.; Mukherjee, A.K. Heat and salinity induced oxidative stress and changes in protein profile in 1120
Amaranthus lividus L. Indian J. plant Physiol. 2006, 11, 41–47.
104. Punia, H.; Madan, S.; Malik, A.; Sethi, S.K. STABILITY ANALYSIS FOR QUALITY ATTRIBUTES IN DURUM 1193
WHEAT (TRITICUM DURUM L.) GENOTYPES. BANGLADESH J. Bot. 2019, 48, 967–972